# CD93 in Health and Disease: Bridging Physiological Functions and Clinical Applications

**DOI:** 10.3390/ijms26178617

**Published:** 2025-09-04

**Authors:** Menghan Cai, Xiaoxi Zhou, Songna Wang, Xuan Huang, Wei Chen, Yiling Chen, Litao Huang, Yan Yan, Yizhun Zhu, Li Ye

**Affiliations:** 1School of Pharmacy and Laboratory of Drug Discovery from Natural Resources and Industrialization, Macau University of Science and Technology, Macau SAR 999078, China; 3220006407@student.must.edu.mo (M.C.); 3230005535@student.must.edu.mo (X.Z.); 3230006875@student.must.edu.mo (S.W.); 21111030066@m.fudan.edu.cn (X.H.); 3230005606@student.must.edu.mo (W.C.); ylchen@must.edu.mo (Y.C.); lihuang@must.edu.mo (L.H.); yanyan@livzon.cn (Y.Y.); yzzhu@must.edu.mo (Y.Z.); 2Department of Biological Medicines, Shanghai Engineering Research Center of Immunotherapeutics, School of Pharmacy, Fudan University, Shanghai 201103, China

**Keywords:** CD93, cancer, tumor angiogenesis, immunotherapy, immune checkpoint inhibitors

## Abstract

CD93 is a highly glycosylated transmembrane glycoprotein with key functional domains, including a C-type lectin-like domain (CTLD) and epidermal growth factor (EGF)-like domains. Primarily expressed in endothelial cells (ECs), CD93 regulates critical physiological processes such as angiogenesis, cell adhesion, migration, and apoptotic cell clearance through interactions with ligands like multimerin-2 (MMRN2) and insulin-like growth factor-binding protein 7 (IGFBP7). Aberrant CD93 expression has been observed in various pathological conditions, including inflammation, cardiovascular diseases, autoimmune disorders, and cancer. Notably, CD93 is overexpressed in tumor-associated blood vessels, which is associated with poor prognosis and advanced disease stages. Targeting the CD93 signaling pathway has the potential to improve tumor vascular function and enhance the efficacy of immunotherapy, making it a promising therapeutic target. This review summarizes the current understanding of CD93’s structure, function, and disease mechanisms, providing a framework for further research and clinical translation in related fields.

## 1. Introduction

CD93, also known as C1qr1, C1qRp, and AA4, is a highly glycosylated type I transmembrane glycoprotein of 652 amino acids [1]. It is encoded by a gene located on chromosome 20’s short arm at region 11.21 in humans and on chromosome 2 in mice [2,3,4]. The CD93 gene includes three single-nucleotide polymorphisms (SNPs): rs2749817, rs2749812, and rs3746731. Specifically, rs2749817 is downstream of the 3′ untranslated region (UTR), rs2749812 is within the 3′ UTR, and rs3746731 is in the first exon. These SNPs may affect the gene’s translation and post-translational modification [5,6,7]. Initially cloned for its role in C1q-triggered phagocytosis, CD93 was termed the complement component C1q receptor (C1qRp). However, further studies have shown that CD93 does not directly bind C1q but interacts with various extracellular matrix proteins in the absence of calcium [2,8].

CD93 belongs to a novel family of transmembrane glycoproteins, the C-type lectin domain (CTLD) group 14 family. This recently discovered CTLD family shares similar domain architectures, binding partners, expression patterns, and functions, and can be secreted or expressed on the cell surface [9]. The extracellular domain architecture and tissue expression profiles of this family resemble those of other adhesion molecules, such as selectins and platelet-endothelial cell adhesion molecules, suggesting their involvement in adhesion and/or leukocyte recruitment processes, which are key functions of the innate immune system [10]. The CTLD family is characterized by the C-type lectin binding domain, an evolutionarily conserved protein superfamily with a double-loop, two-strand anti-parallel β-sheet binding site. The significant sequence variability and unique domain architecture within the CTLD structure allow more than 1000 known mammalian CTLDs to be divided into 17 subfamilies, exhibiting extensive diversity in their domain architectures, signaling pathways, and functions, including inflammation, cell adhesion, carbohydrate recognition, and angiogenesis [11,12,13].

The CTLD family member CD93 is closely related to three other transmembrane glycoproteins: thrombomodulin (CD141, TM), CD248/endosialin (tumor endothelial marker 1/TEM1), and CLEC14a (as shown in Table 1) [9]. CD93 exhibits the closest association with thrombomodulin, as they are located in close proximity (approximately 30,000 nucleotides apart) on chromosome 20 and are highly expressed in ECs, suggesting a potential gene duplication event [14]. TM is a natural anticoagulant and regulator of the innate immune response, while endosialin is a marker of tumor blood vessels involved in regulating tumor growth [3,4].

The CD93 protein is a transmembrane glycoprotein belonging to the CTLD family. Its structural features include an N-terminal signal peptide, a CTLD containing conserved cysteine residues, a sushi-like domain, five epidermal growth factor (EGF)-like domains, a mucin-like region with predicted O-linked glycosylation sites, a transmembrane region, and a cytoplasmic tail (as shown in Figure 1) [2]. The CTLD, initially described as a calcium-dependent carbohydrate-binding domain, is now known to bind a variety of ligands, including proteins, lipids, and inorganic molecules, without requiring calcium [15]. The CTLD adopts a unique “loop-in-loop” folding structure stabilized by a hydrophobic core formed by conserved residues [12]. Proteins containing CTLDs are widely distributed across species and have even been identified in Bordetella bronchiseptica phages [16]. The sushi domain, also known as the complement control protein domain or short consensus repeat, exhibits extensive sequence variation, with its tertiary structure maintained by four conserved cysteine residues forming two disulfide bonds and an invariant tryptophan. This extracellular motif promotes protein–protein interactions [17]. And the structure of the CTLD and sushi domain of CD93 (CTLD-Sushi structural) has been confirmed by PDB deposition under the accession code 8A59 [18]. The EGF-like domain is a conserved structural motif initially described in epidermal growth factor. This module consists of a repeating sequence of 30–40 amino acids, containing six cysteine residues that form three disulfide bonds. EGF-like domains are commonly found in animal proteins, often occurring in tandem repeats, and are widely distributed across diverse protein families [19]. The number of EGF-like subunits can vary within each protein. These domains can exist in either membrane-bound or soluble secreted forms, the latter facilitated by metalloproteinases. The mucin-like domain is characterized by a high content of serine, proline, and threonine residues, enabling extensive post-translational modifications, particularly glycosylation [2]. The abundant O-linked glycans likely confer a more rigid and extended conformation to the protein [20]. Highly glycosylated mucin-like regions are often associated with adhesion proteins, such as CD164, mucosal addressin cell adhesion molecule-1 (MAdCAM-1), and P-selectin glycoprotein ligand-1 (PSGL-1) [21,22]. The C-type lectin-like domain group 14 family members exhibit significantly greater molecular weights than predicted based on their primary amino acid sequences. This discrepancy can be attributed to extensive post-translational modifications, predominantly glycosylation. For instance, the predicted molecular weight of CD93 is 68 kilodaltons (kDa), yet its relative mobility under reducing conditions is 126 kDa. Treatment with an O-linked glycosylation inhibitor can reduce its relative mobility, and enzymatic removal of O-glycosylation decreases its molecular weight [23]. The O-glycosylation of CD93 is crucial for its stable expression on the cell surface. Park et al. demonstrated that inhibiting CD93 glycosylation, either by treating the human U937 cell line with 2-acetamido-2-deoxy-α-D-galactopyranoside benzyl ester or by expressing CD93 in the Chinese hamster ovary (CHO)-K1 cell line (which has a protein glycosylation defect), leads to a decrease in CD93 expression on the cell surface and its detection in the culture medium, indicating that the absence of glycosylation causes rapid release of CD93 into the culture supernatant [24].

## 2. The Expression of CD93

CD93 is primarily expressed by ECs, but its expression has also been observed in various hematopoietic and non-hematopoietic cell types, including hematopoietic stem cells, cytotrophoblasts, monocytes, neutrophils, B cells, natural killer (NK) cells, and platelets [25,26,27,28,29,30]. During embryonic development, widespread CD93 expression was detected in primitive hematopoietic cells, the aorta, gonads, and mesonephros regions of fetal rats. CD93 was first observed in the endocardium and vascular ECs of mouse embryos on day 9, and by day 10, it was uniformly expressed on the ECs of all blood vessels. Shortly after, a small amount of CD93 expression was also detected on hematopoietic cell clusters in the aorta, omphalomesenteric artery, and umbilical artery. As the liver entered its active hematopoietic phase, CD93 expression on vascular and liver ECs decreased, while CD93-positive cells were observed in the liver and large blood vessels. Although CD93 expression decreases in adult humans and mice, its lack during embryonic development does not lead to obvious adverse consequences [31]. CD93 is predominantly expressed on ECs in tissues like the kidney, placenta, spleen, heart, and brain. Studies by McGreal et al. and Dean et al. have identified CD93 on ECs in the heart, frontal cortex, adrenal gland, spleen, kidney, prostate, pancreas, and liver in human autopsy samples. Additionally, CD93 has been detected on ECs in inflamed human tonsils and rat vascular ECs [8,32]. Fonesca et al. used immunofluorescence with custom rabbit anti-peptide polyclonal antibodies to identify CD93 on human umbilical vein endothelial cells (HUVEC) and human monocytes [33]. While CD93 expression is typically low on normal ECs, it increases during embryonic angiogenesis and in pathological conditions such as inflammation and malignancy, aiding in vascular lumen formation, endothelial cell adhesion and migration, and the coordination of surrounding mesenchymal tissue [34,35,36].

CD93 is expressed on hematopoietic stem cells across species. In murine embryos, CD93 expression has been detected in sites of hematopoiesis, including the yolk sac, para-aortic splanchnopleuric mesoderm, and aorta-gonad-mesonephros region [37,38,39,40]. Furthermore, CD93 transcripts have been identified in the adult mouse bone marrow, the primary hematopoietic niche in adults [31]. Consistent with these findings, phenotypic analysis of human cord blood and adult bone marrow cells has revealed CD93 expression on hematopoietic stem cells, defined as lineage-negative CD45^+^ CD38^−^ CD34^+^ cells [28]. However, studies have shown that CD93 alone is not a reliable marker for isolating human hematopoietic stem cells, as other differentiated cells within the hematopoietic compartments also express this molecule [28,41]. Therefore, the precise function of CD93 in hematopoiesis remains to be elucidated.

Lovik et al. developed an anti-CD93 monoclonal antibody, identifying CD93 expression on rat NK cells, iNKT cells, dendritic cells, granulocytes, and activated peritoneal macrophages. The antibody did not affect NK cell cytotoxicity, and CD93 was absent on conventional CD3^+^ NKR^−^ P1^−^ T cells [42].

Ikewaki et al. used a self-produced CD93 monoclonal antibody (mNI-11) with three commercial antibodies (R139, R3, and X-2) to identify a novel cell population (CD4^+^ CD45RA^+^ CD93^+^) in neonatal umbilical cord blood, which may be significant in cell biology, transplantation, and immune responses [29].

Chevrie et al. found that in both T cell-dependent and independent responses, high CD93 expression was confined to antibody-secreting cells, while naive, memory, and germinal center B cells were CD93-negative. CD93-deficient mice could not sustain antibody secretion or maintain bone marrow plasma cells, highlighting CD93’s role in plasma cell maintenance in the bone marrow microenvironment [30]. In 2023, Robinson et al. confirmed the association between CD93 and antibody-secreting plasma cells (ASC), linking high expression to “long-lived” ASC (LL ASC) [43].

Fonseca et al. employed custom antibodies to identify CD93 in vascular ECs, specific brain pyramidal neurons, and neutrophils, but not in most tissue macrophages [33]. This finding is consistent with McGreal et al. and Dean et al., who also reported the absence of CD93 in human tissue macrophages and rat macrophage cell lines [8,32]. In contrast, Norsworthy’s work showed murine CD93 on peritoneal macrophages, particularly those induced by thioglycollate [44]. A 2011 study corroborated CD93 expression in mouse bone marrow-derived macrophages [45]. Despite the ongoing debate over CD93 expression in macrophages, both Fonseca et al. and Norsworthy et al. observed diminished phagocytosis and apoptotic cell clearance in CD93 knockout mouse macrophages, suggesting a role for CD93 in the phagocytosis of cellular debris and dead cells [33,44].

The inability to detect CD93 on certain cell surfaces or the contested expression patterns may result from the cleavage of cell-associated CD93 into its soluble form, a topic to be discussed in detail later.

## 3. Binding Ligands and Interaction Proteins of CD93

CD93, a cell surface protein, mediates various physiological functions by interacting with multiple ligands (as shown in Figure 2).

### 3.1. Moesin

The cytoplasmic domain of CD93 features a positively charged region proximal to the membrane. This structural characteristic enables the interaction of CD93, when localized at the cell periphery, with Moesin, a member of the ERM (Ezrin, Radixin, and Moesin) protein family [46]. This interaction can mediate polarized actin cytoskeleton remodeling through the branching of linear actin fiber bundles [47]. Researchers constructed glutathione S-transferase (GST) fusion proteins containing either the 47-amino acid cytoplasmic domain (Cyto) of GST or various mutants of the CD93 intracellular structural domain. The aim was to identify intracellular molecules that bind to CD93. When cell lysates or recombinant Moesin were used as a source of interacting molecules, the researchers found that the membrane proteins could bind to the GST-Cyto fusion protein. Furthermore, under physiological ionic conditions, the addition of phosphatidylinositol 4,5-bisphosphate facilitates the binding of full-length Moesin to CD93. Interestingly, deleting the last 11 amino acids (C11) from the C-terminus of the CD93 tail significantly enhances its binding to Moesin [46]. Additionally, knockdown of CD93 in ECs results in decreased cell spreading, reduced intercellular adhesion, and disrupted cell–cell contacts, which can be rescued by reintroducing wild-type CD93, but not a mutant lacking the Moesin-binding motif [48]. These findings underscore the critical role of the CD93-Moesin interaction in maintaining endothelial cell adhesion integrity.

### 3.2. GIPC

In 2005, Bohlson et al. conducted a yeast two-hybrid screen to identify proteins binding to the cytoplasmic tail of CD93 [49]. They identified Gα Interacting Protein (GAIP) Interacting Protein C-terminal (GIPC), which regulates migration in various systems and interacts with CD93 via a previously unrecognized CD93 class I PSD-95/Dlg/ZO-1 (PDZ) binding domain in C11 [50,51]. In mammalian cell lysates, the C11 domain and the highly charged near-membrane region of CD93 are essential for its effective interaction with GIPC. Upon stimulation, the C11 domain of CD93’s cytoplasmic tail modulates phagocytic activity by interacting with PDZ domain-containing proteins. Additionally, GIPC interacts with myosin VI, α-actinin, and kinase family member KIF-1B. Both myosin VI and α-actinin are known to regulate endocytosis, highlighting GIPC’s crucial role in this process [46,49]. Another study found that overexpressing wild-type GIPC inhibited HUVEC migration in a damage assay [51].

### 3.3. Multimerin-2 (MMRN2)

The CD93 receptor is primarily activated by the endothelial-specific extracellular matrix (ECM) protein multimerin-2 (MMRN2). Galvagni et al. characterized this interaction and determined the binding affinity using surface plasmon resonance technology [52]. MMRN2, also known as Endoglyx-1, is a member of the elastin microfibril interface proteins (EMILINs) family [53]. Previous studies have demonstrated that the angiogenic effect of MMRN2 binding to CD93 depends on the CTLD of CD93 [52]. Specifically, the interaction involves a 79-amino acid DX domain located between the CTLD and the EGF-like domain repeat sequences, which overlaps with the CTLD [54], as well as the phenylalanine residue at position 238 (F238) in the coiled-coil domain of MMRN2. CD93 primarily interacts with MMRN2 through the DX domain, but the lack of secondary structural elements in this domain compromises its stability. Therefore, the CTLD is essential for maintaining the correct folding of the DX domain [52,55]. Studies have shown that blocking the interaction between CD93 and MMRN2 with monoclonal antibodies (mAbs) targeting the CTLD of CD93 inhibits the adhesion, migration, and lumen formation of HUVEC. The research by Galvagni et al. demonstrated that disrupting the interaction between CD93 and MMRN2 reduced the adhesion and migration of ECs, suggesting that blocking this binding could be a viable treatment for pathological angiogenesis. The CD93-MMRN2 interaction has been shown to play a crucial role in promoting endothelial cell function [52,54]. This interaction regulates the deposition and organization of fibronectin, a process involved in fibrosis formation. During tumor angiogenesis, the CD93-MMRN2 interaction was found to modulate β1 integrin signaling and fibronectin fibril formation in ECs, activating focal adhesion kinase (FAK). Knockdown of MMRN2 or CD93 weakened β1 integrin activation, leading to deformation of the fibronectin network and preventing its organization into a fibrous structure. Additionally, the binding of MMRN2 to CD93 contributes to the stability of CD93 localization in endothelial filopodia by inhibiting the proteolytic cleavage of CD93 [56]. Furthermore, the co-localization of CD93 and MMRN2 expression has been confirmed in the blood vessels of various human solid tumors, including melanoma, Ewing’s sarcoma, ovarian cancer, and glioma [52,56].

### 3.4. Insulin-like Growth Factor Binding Protein 7 (IGFBP7)

The recently discovered glycoprotein IGFBP7, also known as IGFBP-rP1, AGM, T1A12, TAF, mac25, and PSF is a natural ligand of the CD93 receptor. IGFBP7, a 30 kDa member of the IGFBP family, features multiple conserved structural domains, including N-terminal, C-terminal, and central regions. The N-terminal domain harbors the IGFBP motif (GCGCCXXC), crucial for binding insulin-like growth factors (IGFs). IGFBP7 interacts with various molecules such as insulin, activin, and chemokines (e.g., CCL5, CCL21, CXCL10), as well as growth factors like VEGF-A and IGFs. Known receptors for IGFBP7 include the IGF-I receptor, integrin αvβ3, and CD93. Functioning as an extracellular matrix protein, IGFBP7 regulates fundamental biological processes such as proliferation, apoptosis, and migration, and significantly influences tumor development and angiogenesis [57,58]. During physiological angiogenesis, IGFBP7 does not directly stimulate endothelial cell growth or migration, but rather supports efficient adhesion of endothelial cells, activates normal fibroblasts, and induces the expression of junction proteins to promote lumen formation. Moreover, IGFBP7 induces endothelial cell contraction through actin stress fibers and relaxes VE-cadherin-mediated intercellular junctions, thereby affecting vascular permeability and contributing to vascular stabilization and maturation [59,60]. Upregulated IGFBP7 has also been observed in the vascular system of patients with traumatic brain injury and stroke, suggesting its involvement in vascular repair and remodeling processes, such as muscle hypertrophy, wound healing, and physiological and pathological fibrosis [57,61,62]. Additionally, the role of IGFBP7 in cancer has been extensively studied, with several reports confirming its overexpression in tumor vasculature and some human cancer cell lines. IGFBP7 exhibits a dual role in cancer, exerting antitumor effects by inhibiting tumor cell growth and accelerating apoptosis, while also inducing a disordered tumor vasculature system by binding to CD93 [58,63]. Sun et al. revealed that the interaction between CD93 and IGFBP7 is independent of IGF, relying instead on the CTLD structural domain and non-CTLD fragments of CD93 (specifically, the first two epidermal growth factor-like domains) alongside the N-terminal insulin-like growth factor-binding domain of IGFBP7 (PDB ID: 8IVD) [64]. This CD93-IGFBP7 interaction leads to aberrant tumor vascularity. By inhibiting their binding without altering vascular density, pericyte coverage is improved, promoting vascular maturation, reducing vascular leakage, decreasing tumor hypoxia, and enhancing tumor perfusion. These changes facilitate drug delivery and augment the anti-tumor efficacy of chemotherapy and immunotherapy. Additionally, the signal molecules associated with the VEGF and TGF-β pathways, influenced by IGFBP7, may be affected by the IGFBP7-CD93 interaction, although the precise signaling pathway requires further validation. In 2024, further research demonstrated that IGFBP7 and CD93 co-localize on the surface of human aortic endothelial cells (HAECs). IGFBP7 causes endothelial cell damage by interacting with CD93 and reducing SIRT1 expression through an autocrine mechanism [65].

### 3.5. β-dystroglycan (DG)

Proteomic analysis conducted by Galvagni et al. revealed that the cell surface protein CD93 can interact with β-dystroglycan, an extracellular matrix protein that binds laminin and is upregulated on activated ECs. This interaction is facilitated by phosphorylation of the extracellular domain of CD93, which can promote endothelial cell adhesion and migration to the extracellular matrix. Upon binding, the Src kinase phosphorylates tyrosine residues Tyr628 and Tyr644 in the cytoplasmic tail of CD93, stimulating the phosphorylation of downstream effectors such as Cbl [66,67].

### 3.6. Adaptor Protein Cbl

The cytoplasmic domain of CD93 contains a binding motif analogous to that found in the E3 ubiquitin ligase Cbl-binding protein APS, which encompasses a pleckstrin homology domain and an Src homology 2 domain. Co-immunoprecipitation experiments in human umbilical vein ECs confirmed the interaction between CD93 and Cbl. Cbl plays a role in cell adhesion and actin cytoskeleton organization, and this interaction is disrupted by β-dystroglycan knockdown [66,68]. When phosphorylated at Tyr774, Cbl may function as an adaptor protein rather than a ubiquitin ligase, facilitating Crk recruitment and inducing cell migration and tubular structure formation [69].

### 3.7. Vascular Endothelial Cadherin (VE-Cadherin)

VE-cadherin is a transmembrane adhesion protein specifically expressed on the surface of vascular ECs. It primarily mediates cell–cell adhesion between ECs and is involved in the regulation of vascular endothelial growth factor (VEGF) receptor function, endothelial cell permeability, and vascular integrity. Therefore, VE-cadherin is closely associated with angiogenesis [70]. Lugano et al. demonstrated that CD93 can interact with VE-cadherin and limit its phosphorylation and turnover. CD93 deficiency induces VE-cadherin phosphorylation under basal conditions, which disrupts endothelial cell connectivity. In conclusion, CD93 regulates the phosphorylation and turnover of VE-cadherin at endothelial junctions through a Rho/Rho kinase-dependent pathway, with implications for vascular stability and permeability [71]. Additionally, VE-cadherin is a biomarker for vasculogenic mimicry (VM), a new model of tumor microcirculation. Unlike classical tumor angiogenesis, which depends on ECs, VM involves tumor cells exhibiting endothelial-like behavior and forming tubular structures to transport blood, allowing them to obtain oxygen and nutrients independently of normal blood vessels. VM represents a novel model of tumor microcirculation, distinct from classical tumor angiogenesis that relies on ECs. In VM, tumor cells mimic endothelial behavior, forming tubular structures to transport blood, thereby independently accessing oxygen and nutrients without relying on conventional vasculature [72]. First introduced by Maniotis et al. in 1999 in human uveal melanoma, this phenomenon challenges the traditional view that endothelial cell-mediated angiogenesis is the sole mechanism for tumor growth and metastasis, while also complementing existing angiogenesis theories [73]. Recently, VM has been identified in various malignancies, including glioblastoma, hepatocellular carcinoma, breast cancer, and lung cancer. VM correlates with tumor malignancy, growth, progression, metastasis, invasion, durg resistance, and poor prognosis [70,72]. In 2001, Hendrix et al. demonstrated that reducing VE-cadherin expression in invasive melanoma cells disrupted VM network formation, directly linking VE-cadherin to VM. [74] Similar findings were observed in a pancreatic cancer model [75]. VE-cadherin primarily modulates the phosphorylation and localization of embryonic stem cell markers, such as EphA2, to activate FAK and extracellular signal-regulated kinase (Erk). Independently, activated EphA2 can also initiate the PI3K/MMP pathway, promoting the formation of VM [76]. On the other hand, CD93 induces VE-cadherin phosphorylation, disrupting endothelial cell junctions and the vascular barrier, leading to the leakage of plasma proteins (e.g., fibrinogen) and the formation of a fibrous network, which is a key factor in VM [71]. Moreover, knockdown of CD93 and MMRN2 inhibits the activation of integrin β1, thereby suppressing the PI3K/AKT/SP2 signaling pathway and inhibiting VM formation [77].

### 3.8. Interleukin-17D (IL-17D) 

Interleukin-17D, a member of the interleukin-17 cytokine family, has been detected in rheumatoid nodules and psoriatic skin lesions, suggesting a significant role in inflammatory disorders [78,79]. This cytokine expressed across various tissues, including the brain, skeletal muscle, adipose tissue, pancreas, lung, colon, and heart. Recent work by Huang et al. in 2021 identified the cell surface CD93 as a functional binding partner for IL-17D, demonstrating that this interaction regulates type 3 innate lymphoid cells (ILC3) and contributes to intestinal homeostasis [80]. Additionally, the IL-17D-CD93 axis helps maintain the integrity of the intestinal epithelial barrier by reducing inflammation and barrier damage [81]. Furthermore, studies have implicated the IL-17D signaling pathway in the pathogenesis of skin inflammation (e.g., psoriasis) [82], neurological inflammation [83] and cardiovascular inflammation [84,85].

Additionally, the recombinant extracellular domain of CD93 has been shown to bind to the surface of THP-1 cells, suggesting the existence of an uncharacterized CD93 ligand in this monocytic cell line, warranting further investigation [86].

## 4. Physiological Function

In human umbilical vein ECs, RNA interference silencing of CD93 impairs cell proliferation, migration, adhesion, and sprout formation [54]. CD93 inherently promotes efferocytosis, cell adhesion and migration, angiogenesis, and maturation.

### 4.1. Adhesion and Migration

Integrins are a class of proteins that play a crucial role in connecting the cytoskeleton and signaling pathways [87]. They are involved in the activation of numerous signaling proteins, including FAK, Src, and Cas [88]. Cell–matrix interactions trigger integrin-mediated signaling, leading to the phosphorylation dependent activation of Src during CD93 phosphorylation. Dystroglycan exists in a dynamic regulatory state within ECs. Phosphorylated DG recruits and activates Src, which then phosphorylates the cytoplasmic tail of CD93. This facilitates the binding of the adaptor protein Cbl, which can regulate cell migration [66,67], and creates a binding site for Crk, thereby recruiting Crk. Crk is an activator of the Rho/Rac family of small GTPases and is involved in the regulation of intracellular actin dynamics [69]. Crk interacts with DOCK180 to modulate the activity of Rho GTPases, contributing to cytoskeletal remodeling [89]. At the leading edge of migrating cells, CD93 specifically enhances the activity of Rac1 and Cdc42 while reducing the activity of RhoA [71,89]. Rho GTPases, such as RhoA, Rac1, and Cdc42, constitute a pivotal class of cell signaling proteins within the Ras superfamily. In conjunction with CD93, Cbl, and Crk, they orchestrate cytoskeletal remodeling, a process integral to endothelial cell adhesion and migration. Consequently, Rho GTPases are crucial regulators of cell motility [67,90]. The intracellular domain of β-dystroglycan interacts with dystrophin, which associates with the actin cytoskeleton [91]. Downregulation of actin stress fiber formation in ECs impairs cell adhesion and cell–cell contacts. The monoclonal antibody mNI-11, which recognizes CD93, can promote the adhesion and spreading of HUVEC. The use of microtubule inhibitors and calcium chelators can eliminate the mNI-11-induced spreading of HUVEC, indicating the importance of calcium ions and the cytoskeleton in regulating cell adhesion [92].

Galvagni et al. demonstrated that during the initial stages of endothelial cell spreading, CD93 localizes to the apical lamellipodia, concomitant with the localization of activated moesin, vascular endothelial cadherin, and Par3 protein in the apical region [46]. Importantly, this apical localization is dependent on the actin cytoskeleton, as it can be eliminated by moesin silencing or disruption of the actin network [47]. Barbera et al. further elucidated the underlying mechanism, revealing that moesin and F-actin are crucial for the recycling of CD93 in adherent and migrating ECs. Specifically, the small GTPase Rab5c drives the recycling of CD93 back to the endothelial cell surface. Within the Rab5c endosomal compartment, CD93 forms a complex with Multimerin-2 and active β1 integrin, which is then recycled back to the basolateral, polarized cell surface through clathrin-independent endocytosis, ensuring a continuous supply of active signaling components required for endothelial cell adhesion and migration [93].

### 4.2. Angiogenesis and Vascular Maturation

The expression of cell adhesion molecules on endothelial surfaces is crucial for angiogenesis, supporting the growth and survival of new blood vessels [34,94]. Galvagni et al. demonstrated that silencing the CD93 gene in ECs elevates dystroglycan expression, a molecule implicated in angiogenesis and overexpressed in the vascular endothelium of malignant tumors. They found that the interaction between β-DG and CD93 enhances endothelial cell migration and orchestrates the formation of capillary-like structures [66].

Multiple studies have firmly established the role of CD93 in angiogenesis. A monoclonal antibody targeting the interface between the C-type lectin-like domain and sushi domains of CD93 (clone 4E1) exhibited anti-angiogenic activity in both in vitro and in vivo experiments, further emphasizing the importance of CD93 in this process. Blocking VEGF with bevacizumab led to downregulation of the CD93 gene [2], and the inhibition of vascular endothelial growth factor receptor 2 (VEGFR2) and fibroblast growth factor 1 (FGF1) by brivanib alaninate significantly reduced CD93 expression [95]. Notably, treating ECs with recombinant CD93 protein promoted lumen formation and sprouting, accelerated wound healing, and induced the formation of blood vessel-like structures in the subcutaneous tissue of mice. Conversely, wound repair was delayed in CD93 knockout (CD93^−/−^) mice due to impaired neovascularization [96]. Kao et al. generated a recombinant soluble CD93 protein comprising the mucin domain, EGF repeats, and C-type lectin-like domain, which induced the differentiation of HUVECs through the phosphoinositide 3-kinase/Akt/endothelial nitric oxide synthase and extracellular signal-regulated kinase-1/2 signaling pathways in in vitro angiogenesis experiments [97].

Vascular permeability is central to tissue homeostasis, with endothelial cell junction regulating molecular transport between blood and tissues [98]. Downregulation of CD93 disrupts VE-cadherin in endothelial junctions, increasing vascular permeability. Specifically, the absence of CD93 in ECs and in CD93 knockout (CD93^−/−^) mice induces phosphorylation of VE-cadherin, leading to its internalization and an increase in blood–brain barrier permeability.

### 4.3. Clearance of Apoptotic Cells

Phagocytosis is a critical process in maintaining tissue homeostasis, as it can prevent inflammatory and autoimmune diseases triggered by the lysis of apoptotic cells. This process facilitates the clearance of apoptotic cells, thereby avoiding inflammation [86]. CD93 plays a pivotal role in this process. A 2004 study by Norsworthy et al. found that CD93-deficient mice exhibited a reduced ability to clear apoptotic cells in vivo. When apoptotic cells were injected into the inflamed peritoneal cavity, macrophages from CD93^−/−^ mice phagocytosed significantly fewer apoptotic Jurkat cells or thymocytes compared to those from age-matched control mice [44]. Additionally, a study using the THP-1 cell line, as well as human monocytes and macrophages, demonstrated that soluble CD93 (sCD93) binds to apoptotic cells through its CTLD and effectively opsonizes them by binding to the integrin αxβ2 via its EGF-like repeats, whereas membrane-bound CD93 does not exhibit phagocytic or efferocytic activity. Therefore, sCD93 acts as an opsonin that can bind to apoptotic cells and the phagocytic integrin subunit αxβ2 on macrophages, although CD93 itself does not affect the phagocytic activity of macrophages, it may contribute to the clearance of dying cells in vivo [86].

### 4.4. Stabilized Thrombin Receptor

Trivigno et al. utilized flow cytometry and light transmission aggregometry to demonstrate that in the absence of CD93, platelet aggregation induced by thrombin receptor PAR4 stimulation was significantly impaired. This defect was associated with reduced α-granule secretion, integrin αIIbβ3 activation, and protein kinase C (PKC) stimulation. Wild-type (WT) and CD93-deficient platelets exhibited comparable levels of PAR4 expression in the resting state. However, following stimulation with the PAR4-activating peptide, PAR4 clearance from the platelet surface was more pronounced in CD93-deficient platelets compared to WT controls. These findings indicate that CD93 is essential for the stable expression of thrombin receptors on the cell surface, thereby supporting platelet activation triggered by PAR4 stimulation [99].

## 5. Soluble CD93

CD93 exists on the cell surface in both membrane-bound and soluble forms. Specific stimuli can cleave its extracellular domain, prodncing sCD93 detectable in human plasma. This soluble form is crucial as an angiogenic factor [97], a diagnostic and prognostic marker in metabolic and inflammatory diseases [100,101], and in mediating macrophage recruitment and phagocytosis [15,102].

The study by Bohlson et al. in 2005 demonstrated reported that stimulation of monocytes and neutrophils with phorbol 12,13-dibutyrate (PDBu), lipopolysaccharide (LPS), and tumor necrosis factor-α (TNF-α) resulted in the shedding of the extracellular domain of CD93, releasing a soluble fragment [103]. This soluble fragment retains the N-terminal carbohydrate recognition domain and EGF repeats, which can exert angiogenic functions in ECs through the EGF-like domain [103]. While the extracellular domain of full-length CD93 can also induce these angiogenic signals, the construct lacking the CTLD triggers a more significant effect by enhancing the EGFR1-mediated PI3K signaling pathway [104]. Similarly to CD44, the shedding of CD93 may involve metalloproteinases, including members of the a disintegrin and metalloproteinase (ADAM) family members, particularly TNF-α converting enzyme (TACE, ADAM17), matrix metalloproteinases (MMPs), and soluble neutrophil-derived serine proteases [105]. The shedding of CD93 can be inhibited by the metalloproteinase inhibitor 1,10-phenanthroline, but is independent of ADAM17 [103]. Furthermore, the lack of O-linked glycosylation within the mucin-like domain of CD93 or knockdown of MMRN2 can enhance the shedding of CD93 from the cell membrane, thereby increasing the level of soluble CD93 in the culture medium [24].

Neutrophils exhibit a dual response to the phorbol ester PDBu, upregulating the surface expression of CD93 while also inducing its shedding. The intracellularly stored CD93 rapidly replaces the shed CD93 on the neutrophil surface. Subsequent in vivo experiments confirmed this inflammation-induced CD93 shedding. PDBu can also stimulate the release of sCD93 from thioglycolate-induced peritoneal macrophages in mice. Compared to PBS-treated controls, C57BL/6 mice receiving thioglycolate showed significantly elevated levels of sCD93 in the serum (1.9-fold) and peritoneal lavage fluid (4.2-fold) [106]. In conclusion, sCD93 can serve as a marker of inflammation.

SCD93 plays a crucial role in efferocytosis, the phagocytic clearance of apoptotic cells. Specifically, sCD93 binds to the apoptotic cells in a calcium-independent manner through its CTLD, acting as an opsonin to facilitate phagocytosis by macrophages. This interaction is mediated by the binding of sCD93’s EGF repeats to the αxβ2 integrin on the macrophage surface [86].

The level of soluble CD93 has been directly linked to various disease states. For instance, elevated sCD93 levels have been reported in the synovial fluid of patients with rheumatoid arthritis [86]. Moreover, the plasma concentration of sCD93 has been proposed as a biomarker for conditions such as systemic sclerosis [101], allergic asthma [107], glucose metabolism regulation [108], and coronary artery disease [6] (the specific role of CD93 in these diseases will be discussed in detail later).

## 6. The Role of CD93 in Diseases

### 6.1. Inflammation

Inflammation is the body’s defense mechanism against pathogens, infectious agents, or tissue damage, involving coordinated interactions between immune cells and blood vessels. This process generates complex molecular signals that activate or modify immune and other cells [109]. Key features include acute microvascular responses, edema, and the recruitment of neutrophils and monocytes. Chemokine gradients produced by cells under inflammatory stimuli guide migratory leukocytes to the damage site, regulated by adhesion events between leukocytes and ECs [110]. Inflammatory effector cells include macrophages, monocytes, neutrophils, eosinophils, basophils, and mast cells [111]. Endothelial growth factors from activated leukocytes stimulate angiogenesis, providing nutrients to the inflamed site and sustaining the lesion [112]. While inflammation is essential for tissue homeostasis, excessive or chronic inflammation can lead to pathological conditions.

Xiao et al. showed that the monoclonal anti-CD93 antibody R3 induces IL-8 production in human umbilical vein ECs [113]. Soluble CD93 triggers IL-6, IL-1β, and TNF-α production, facilitating monocyte differentiation into macrophage-like cells [34]. Wang et al. found that neutrophils from hypoxic subjects had increased CD93 expression upon Escherichia coli stimulation, which was inhibited by a 1 h vitamin E pretreatment before hypoxia induction. In vitro studies revealed that CD93 expression on neutrophils rises upon activation, indicating its presence in granules. These findings confirm that, CD93 expression is upregulated on activated neutrophils in vivo, and antioxidant treatment can inhibit this upregulation or degranulation [103,114]. Adamczyk et al. emphasized the role of plasma-derived extracellular vesicles (pEVs) in modulating inflammation by inhibiting pathogen-associated molecular pattern (PAMP)-induced macrophage activation. This mechanism supports tissue repair functions in macrophages and upregulates the CD93 gene, essential for cell growth and tissue remodeling [115]. Concurrently, another study demonstrated the production of soluble recombinant human CD93 (rCD93) using a mammalian expression system. This study confirmed that soluble rCD93 specifically binds to the pro-inflammatory high-mobility group box 1 (HMGB1) via its lectin-like domain, disrupting the HMGB1-receptor interaction and reducing inflammation. These findings underscore the therapeutic potential of soluble recombinant CD93, particularly the variants containing the lectin-like domain, in managing inflammatory conditions [116].

Asthma, a prevalent chronic inflammatory airway disease, is characterized by recurring symptoms such as wheezing, shortness of breath, chest tightness, and coughing. The condition is associated with heightened cellular inflammation, both eosinophilic-dependent and eosinophilic-independent, as well as overproduction of reactive oxygen species (ROS) [40]. In 2009, Baines et al. reported a correlation between CD93 and asthma. They analyzed whole-genome gene expression profiles from induced sputum samples of participants exhibiting a >10% change in sputum neutrophils following a 14-day low-antioxidant diet, which depletes antioxidants, increases neutrophilic airway inflammation, and worsens asthma. The researchers identified upregulated genes, including CD93, which displayed a 1.5-fold increase in mRNA level [117]. Clinical asthma is primarily categorized into two main phenotypes: allergic asthma (AA) and non-allergic asthma (NA) [118]. In 2015, Raedler et al. observed that NA patients had a notably higher neutrophil count in their blood compared to AA patients. NA patients exhibited elevated expression of pro-inflammatory IL-1β, anti-inflammatory IL-37, and PSTPIP2, along with increased expression of neutrophil-related genes CD93, RGS13, and TREM1 [119]. Subsequently, Sigari et al. reported a significant correlation between sCD93 and asthma, observing elevated serum sCD93 levels in individuals experiencing asthma attacks that subsequently decreased following treatment [120]. These findings were supported by the work of Park et al., who conducted three studies demonstrating that increased serum sCD93 levels were associated with the worsening of various allergic conditions, including allergic rhinitis (AR), chronic spontaneous urticaria (CSU), and asthma [121]. Further investigations by Park et al. using mouse models of acute asthma and lipopolysaccharide-induced airway inflammation revealed a decrease in CD93 levels in lung homogenates and respiratory epithelial cells, alongside an increase in serum sCD93 levels, reinforcing the potential of sCD93 as a biomarker for allergic asthma [122]. In a subsequent 2020 study, Park et al. conducted in vitro and in vivo experiments using dust mite stimulation, observing elevated sCD93 levels in the culture medium and mouse serum. Additionally, a retrospective analysis of 96 human samples indicated significantly higher sCD93 levels in asthma patients compared to healthy controls with sCD93 demonstrating moderate sensitivity (71.4%) and specificity (82.4%) in predicting asthma diagnosis [107].

In the nervous system, Harhausen et al. modeled cerebral ischemia–reperfusion in mice by occluding the middle cerebral artery for 30 to 60 min, followed by euthanizing the mice 48 to 72 h post-reperfusion. They found increased CD93 mRNA and protein levels in ischemic brain tissue compared to non-ischemic tissue in wild-type mice. CD93 knockout mice exhibited heightened inflammatory responses, greater leukocyte infiltration, and elevated chemokines such as CCL21/Exodus-2 after ischemia–reperfusion injury, suggesting CD93’s role in modulating post-traumatic inflammation [123]. In a 2014 study, neuroinflammation was induced by injecting lipopolysaccharide into rat brain ventricles, causing an initial increase and subsequent decrease in CD93 expression on cerebral cortical cell membranes, with distinct cytoplasmic and nuclear staining patterns. This study also confirmed the interaction between CD93 and GIPC, and the rapid increase in CD93 expression post-LPS treatment suggested a potential role or CD93 in early central nervous system inflammation [26]. In 2018, Griffiths et al. established two models of autoimmune encephalomyelitis in wild-type and CD93 knockout mice. Their findings revealed high CD93 expression in neurons, ECs, and microglia, but not in astrocytes or oligodendrocytes. Compared to wild-type mice, CD93-deficient mice with autoimmune encephalomyelitis exhibited heightened brain and spinal cord inflammation, as well as more severe neuronal damage, including increased blood–brain barrier disruption and leakage, in both models [124]. In 2021, Zhong et al. established a mouse model of unilateral brachial plexus root avulsion (BPRA) and observed significant alterations in CD93 levels through cytokine analysis. Following injury, a shift in CD93 expression from the cytoplasm to the nucleus was occasionally observed in neurons within the ipsilateral spinal cord segments [125]. Subsequently, in 2022, Qiao et al. reported a notable upregulation of CD93 and integrin β1 expression in a rat model of pneumococcal meningitis. The interaction between integrin β1 and CD93 facilitated the polarization of microglia towards the M1 phenotype, resulting in increased secretion of pro-inflammatory factors and subsequent neurological damage [83]. Cerebral cavernous malformation (CCM) is a condition characterized by focal neurological deficits, seizures, and an elevated risk of hemorrhagic stroke, displaying inflammatory features. In 2024, Jauhiainen et al. conducted a proteomic analysis of human CCM tissue samples and validated the functional relevance of the marker CD93 in the context of endothelial cell activation and extracellular matrix remodeling using a preclinical mouse model of CCM [126].

In the digestive system, sCD93 is recognized for its crucial role in the phagocytosis of apoptotic cells. Notably, a study inducing peritonitis in mice reported a significant elevation of sCD93 levels in both mouse serum and peritoneal lavage fluid. Greenlee-Wacker et al. demonstrated that CD93 is involved in regulating leukocyte migration and C1q hemolytic activity in thioglycollate-induced peritonitis. Specifically, CD93 knockout mice exhibited a 22% and 46% reduction in C1q hemolytic activity at 1 and 3 h post thioglycollate injection, respectively, accompanied by heightened leukocyte infiltration and compromised vascular integrity. In bone marrow chimeric mice, the restoration of leukocyte recruitment and C1q hemolytic activity was observed when CD93 was expressed on non-hematopoietic cells, and the increase in sCD93 levels in the inflammatory fluid was solely detected in this condition. These findings suggest that membrane-bound CD93, rather than sCD93, is pivotal in modulating leukocyte recruitment and complement activation during mouse peritonitis, indicating its role in reinstating a normal inflammatory response [45].

Finally, in the study of dental inflammation, Li et al. identified 19 core genes associated with periodontitis through extensive analysis of public databases. Among these, CD93 emerged as a key gene, validated for its significant association with periodontitis. The research established a causal link between CD93 and the disease, highlighting its crucial role in the condition’s progression [127].

### 6.2. Cardiovascular Diseases (CVD)

Atherosclerosis a hallmark of cardiovascular disorders, is characterized by endothelial dysfunction, impaired angiogenesis, and inflammation, which significantly contribute to the development and progression of atherosclerotic thrombosis [128,129]. The essential physiological functions of CD93 are integral to the initiation and advancement of atherosclerosis. Notably, the capacity of macrophages to eliminate apoptotic cells is crucial for plaque formation, size, and stability. Consequently, CD93 has garnered increasing attention in cardiovascular research due to its pivotal role. Numerous studies have identified a correlation between CD93 protein levels or gene polymorphisms and the prevalence of cardiovascular risk factors (e.g., hypertension, dyslipidemia, obesity) and diseases (e.g., heart failure, coronary artery disease, ischemic stroke). A comprehensive review by Piani et al. in 2023 offers an in-depth exploration of this topic [130].

Type 2 diabetes involves complex metabolic and inflammatory processes. Research highlights the role of CD93 in glucose metabolism regulation. A 2014 genome-wide association study by Chan et al. examined 8155 Black, 3494 Hispanic American, and 3697 non-Hispanic white American women from the National Women’s Health Initiative and the Genomics and Randomized Trials Network. CD93 was identified among the top 10 biological pathways and regulatory genes linked to cardiovascular disease and type 2 diabetes [131]. Similarly, Strawbridge et al.’s cross-sectional study of 901 individuals with type 2 diabetes and over 2000 controls found significantly lower sCD93 levels in diabetic individuals (*p* < 0.0001). A prospective cohort study further showed that reduced sCD93 levels preceded diabetes onset. In CD93-deficient mice subjected to a high-fat diet, there was decreased glucose clearance and insulin sensitivity, along with pancreatic vascular leakage, compared to controls. However, no significant link was found between sCD93 levels and carotid intima-media thickness (IMT), a marker of atherosclerosis. Additionally, genetic variants associated with diabetes risk, identified by the DIAGRAM Consortium, did not affect sCD93 levels, whether individually or as part of a single nucleotide polymorphism score [108]. Elevated serum levels of sCD93 have been associated with increased urinary albumin-creatinine ratio, decreased estimated glomerular filtration rate (eGFR), and heightened risk of diabetic nephropathy, suggesting its potential utility as a prognostic indicator for this condition [100]. Additionally, studies in a rat diabetic wound model have demonstrated that CD93-positive hematopoietic stem cells, when administered concurrently with polycaprolactone-gelatin nanofiber scaffolds significantly enhanced the therapeutic efficacy for diabetic wounds [132]. More recently, Xu et al. in 2023 corroborated these findings, confirming the beneficial impact of CD93 on wound healing in diabetic mice through the promotion of angiogenesis and re-epithelialization [96].

Macrophages play a crucial role in regulating inflammation and the early stages of atherosclerosis by modulating lipid accumulation in the arterial intima [133]. Cholesterol 27-hydroxylase, expressed by arterial ECs and monocytes/macrophages, serves as a primary defense mechanism against atherosclerosis development. The absence of cholesterol 27-hydroxylase results in the accumulation of oxidized low-density lipoprotein (LDL), anti-LDL antibodies, and foam cells in the arterial wall, contributing to atherosclerotic plaque formation. The inhibition of cholesterol 27-hydroxylase downregulation induced by IC-C1q can be achieved by anti-CD93 antibodies [134]. A study by Van der Net et al. in 2008 revealed that the CD93 gene polymorphism (rs37467) could serve as a predictive marker for coronary heart disease development in patients with familial hypercholesterolemia (FH) [5]. In a recent animal study conducted in 2022, Su et al. established an atherosclerotic model in rats by clamping the left carotid artery. They discovered that adoptively transferred CD93 highly expressing macrophages labeled with 3H-2-deoxyglucose (3H-2-DG) and the injection of 125I-α-CD93 (monoclonal anti-CD93 antibody labeled with 125I) could effectively target CD93 within atherosclerotic plaques suggesting that CD93 could be a promising target for imaging atherosclerotic plaques [135]. A study demonstrated a correlation between CD93-expressing monocyte phenotypes. precursors of M2 macrophages, and apolipoprotein B (ApoB) levels. In non-classical monocytes, CD93 expression inversely correlated with the ApoA1/ApoB ratio. Conversely, in intermediate monocytes, it positively correlated with the total cholesterol/high density lipoprotein cholesterol (HDL-C) ratio and negatively with HDL-C levels [130]. Supporting this, another study identified CD93 as a functional IL-17D receptor in ECs. Inhibition of miR-181a-5p increased cell viability and glutathione levels while reducing reactive oxygen species and IL-6/IL-8 levels. IL-17D induced ferroptosis via the CD93/miR-181a-5p/SLC7A11 pathway, exacerbating endothelial inflammation. These findings suggest CD93 as a potential therapeutic target for atherosclerosis [85]. Additionally, recent research reported significant upregulation of CD93 in plaque tissues compared to non-plaque tissues in peripheral vascular atherosclerosi [136]. Together, these studies underscore the strong link between CD93 expression and atherosclerosis [137].

Lee et al. argued that sCD93 levels do not significantly influence human obesity as measured by body mass index (BMI). In contrast, Strawbridge et al. and Snelder et al. suggested a potential interaction between CD93 and obesity. A study involving individuals with type 2 diabetes and a control group confirmed a positive correlation between sCD93 levels and both adiponectin and vitamin D levels, with an inverse correlation to BMI [108]. Additionally, research on 72 obese patients showed a significant reduction in sCD93 levels one year after bariatric surgery [138]. Several studies have highlighted a significant association between CD93 polymorphism and arterial hypertension. A longitudinal study found a positive link between serum sCD93 and increased arterial hypertension prevalence in the elderly [139]. Van der Net et al. reported a significant association between the rs3746731 polymorphism of the CD93 gene and hypertension (*p* = 0.02) [5]. Moreover, SNPs rs7492 and rs2749812 of CD93 were associated with hypertension in high-altitude areas. Patients with pulmonary arterial hypertension and left heart failure had higher serum sCD93 levels compared to healthy controls (*p* < 0.001) [140].

In 2008, Van der Net et al. identified a single nucleotide polymorphism at amino acid position 541 in the CD93 gene, resulting in a proline to serine missense mutation (rs3746731), has been associated with a 1.26-fold increased risk of coronary heart disease (CHD) [5]. Chandramouli et al. supported this by showing demonstrating a correlation between sCD93 levels and coronary microvascular dysfunction in males with heart failure with preserved ejection fraction [141]. However, the link between CD93 and myocardial infarction remains contentious. While Malarstig et al. reported no significant differences in myocardial infarction risks among various sCD93 tertiles [6], Youn et al. observed significantly elevated sCD93 levels in patients with acute myocardial infarction compared to controls in a case–control study involving 120 patients and 120 matched controls (*p* < 0.0001) [142].

In a study of 33 ischemic stroke patients, CD93 transcript levels were found to be double those of to the control group [143]. Another investigation revealed that elevated sCD93 levels were correlated with a nearly fourfold increase in 90-day mortality risk postacute ischemic stroke [144]. Recent research has identified CD93 as a promising therapeutic target for evaluating functional outcomes after ischemic stroke. Zhang et al. employed drug-target Mendelian randomization, Steiger filtering, and colocalization analyses, identifying CD93 among 16 of over 260 druggable genes causally associated linked to poor functional prognosis in ischemic stroke [145]. In 2009, Bouwens et al. observed a positive correlation between serum sCD93 levels and the hazard ratio(HR) in 263 chronic heart failure patients [146]. Alehagen et al. found that higher plasma concentrations of the CD93 gene polymorphism rs2749812 were associated with increased N-terminal pro-B-type natriuretic peptide (NT-proBNP) levels, a marker of heart failure severity [139].

Finally, patients with left-heart failure complicated by pulmonary hypertension who underwent heart transplantation exhibited postoperative sCD93 levels comparable to those of healthy controls, as reported in the case–control study by Helleberg et al. Conversely, patients with the same condition who did not undergo transplantation demonstrated elevated sCD93 levels compared to healthy controls [130].

Additionally, the study by Tian et al. revealed that the expansion of in ILC3 cells in viral myocarditis(VMC) induced cardiac tissue injury through the activation of S1P/S1PR1 and CXCR6/CXCL16 signaling cascades, suggesting a potential role for the IL-17D-CD93 axis in cardiac inflammation [84].

However, the current literature lacks reports on the relationship between CD93 and other cardiovascular-related pathologies, such as hyperuricemia and peripheral arterial disease [130].

### 6.3. Autoimmune Disorders

Rheumatoid arthritis (RA) is an autoimmune disorder characterized by synovial inflammation, potentially leading to joint deformities. While its etiology remains under investigation, RA is thought to be closely associated with autoimmunity, genetics, and microbial infections. Jeon et al. reported significantly elevated levels of sCD93 in the synovial fluid of RA patients compared to those with osteoarthritis, though the role of CD93 or sCD93 in RA pathogenesis and its correlation with disease severity require further study [86]. CD248 and CD93 are both members of the Group 14 transmembrane glycoprotein family and share structural homology. Payet et al. examined the regulation of CD248 expression in vitro using skin and synovial mesenchymal stem cell lines, as well as samples from RA and osteoarthritis patients. Their findings suggest that both membrane-bound and soluble CD248, acting as a decoy receptor, may contribute to RA pathogenesis. In RA synovial tissues, CD90-positive perivascular mesenchymal stem cells (MSCs) exhibited co-expression of CD248 and VEGF, with high sCD248 concentrations detected in the synovial fluid of RA patients [147].

The CD93 gene has been implicated in the pathogenesis of psoriasis, a chronic inflammatory skin condition. Several studies have investigated the role of CD93 in this disease. Duvetorp et al. reported a significant association between the rs2749817 CD93 gene polymorphism and psoriasis. They found increased CD93 immunofluorescence in the dermal ECs of lesional skin and elevated CD93 gene expression in psoriasis patients compared to controls, irrespective of skin damage. Notably, narrow-band ultraviolet B (NB-UVB) treatment did not alter CD93 gene expression [148]. Shehata et al. corroborated these findings, demonstrating significantly higher positive expression of CD93 in dermal endothelial and inflammatory cells in psoriasis patients compared to controls. They also observed lower frequencies of the T/C and C/C genotypes of the CD93 gene single-nucleotide polymorphism rs2749817 in psoriasis patients, with the C/C genotype present only in some cases [149]. Additionally, Ni et al. reported that the cytokine IL-17D could trigger the activation of the CD93-p38 MAPK-AKT-SMAD2/3 signaling cascade in keratinocytes, leading to the inhibition of DDX5 expression and exacerbating psoriasis-like skin inflammation [82]. In 2023, Liu et al. provided further evidence for the potential of CD93 as a diagnostic marker in psoriasis, elucidating the regulatory mechanisms underlying the disease [150]. These studies collectively suggest that CD93 plays a crucial role in the pathogenesis of psoriasis, with its altered expression and genetic variations contributing to the development and progression of the condition. The involvement of the CD93-mediated signaling pathway in keratinocyte dysfunction highlights the importance of this gene in the pathophysiology of psoriasis.

Systemic lupus erythematosus (SLE) is a chronic, multisystem autoimmune disorder that predominantly affects young women. Apoptotic cells are considered a source of autoantigens in lupus, and the receptor CD93 can effectively opsonize these cells. In a 2006 study, Moosig employed flow cytometry to analyze CD93 expression in 36 SLE patients and 20 healthy controls. Additionally, monocytes from 5 patients and 5 controls were cultured and stimulated with dexamethasone, interferon-γ, and LPS, respectively, to assess CD93 expression. While no significant difference in CD93 expression was observed between SLE patients and healthy controls, in vitro experiments demonstrated that dexamethasone had no effect on CD93 expression. However, CD93 expression was significantly upregulated in patients who received low-dose or no dexamethasone in vivo, or were stimulated with LPS [151].

Systemic sclerosis (SSc), or scleroderma, is a systemic autoimmune disorder characterized by localized or widespread skin thickening and fibrosis [152]. Yanaba et al. reported robust CD93 immunostaining on ECs in the lesional skin tissues of SSc patients. Compared to healthy controls, SSc patients exhibited elevated serum levels of sCD93 (*p* < 0.001). Furthermore, sCD93 levels were higher in patients with diffuse cutaneous SSc than in those with limited cutaneous SSc or systemic lupus erythematosus, suggesting a correlation with the severity of skin sclerosis [101].

### 6.4. Cancer

The growth and metastatic progression of primary tumors rely on the circulatory system to obtain oxygen and nutrients [153]. Tumors can either utilize existing blood vessels or generate new ones through angiogenesis [154]. Consequently, targeting tumor angiogenesis to cut off the tumor’s nutrient supply has been explored as an anti-tumor strategy [155]. CD93 is primarily expressed on ECs and plays a crucial role in angiogenesis and vascular maturation, but its expression is extremely low or undetectable in normal adult tissues of humans and mice. In 2013, Masiero et al. identified the angiogenic/vasculogenic gene signatures in human tumor samples from different cancer types, including head and neck squamous cell carcinomas (HNSCCs), breast cancers (BCs), and clear cell renal cell carcinomas (CCRCCs), and found that CD93 was significantly upregulated on tumor-associated ECs and was one of the top 20 genes associated with tumor angiogenesis [156]. Subsequent studies have shown that CD93 is a downstream effector of VEGF, a potent tumor-derived angiogenesis stimulator, and its expression is significantly downregulated when VEGF is inhibited [63]. Therefore, CD93 has emerged as a novel angiogenesis activator, primarily enhancing endothelial cell adhesion and expediting tumor angiogenesis. Its elevated expression on vascular ECs correlates with accelerated tumor growth and diminished host survival [48]. Furthermore, CD93 is intricately linked to immune cell infiltration and immunotherapy, potentially acting as a new immune checkpoint within the tumor microenvironment (TME) [56,157,158].

Between 2021 and 2022, Tong, Guo, and their team examined CD93 expression across various cancers using multiple public databases. They found notable differences in CD93 expression between tumors and adjacent normal tissues. Elevated CD93 levels were associated with tumor angiogenesis, immune cell infiltration, poor prognosis, and advanced TNM stages [159,160]. Guo et al. reported significant CD93 overexpression in glioblastoma multiforme (GBM), pancreatic adenocarcinoma (PAAD), stomach adenocarcinoma (STAD), cholangiocarcinoma (CHOL), brain lower grade glioma (LGG), liver hepatocellular carcinoma (LIHC), kidney renal clear cell carcinoma (KIRC), acute myeloid leukemia (LAML), HNSCCs, testicular germ cell tumors (TGCT), and skin cutaneous melanoma (SKCM). Conversely, CD93 expression was reduced in colon adenocarcinoma (COAD), kidney renal papillary cell carcinoma (KIRP), kidney chromophobe (KICH), adrenocortical carcinoma (ACC), cervical squamous cell carcinoma and endocervical adenocarcinoma (CESC), uterine corpus endometrial carcinoma (UCEC), uterine carcinosarcoma (UCS), breast invasive carcinoma (BRCA), lung adenocarcinoma (LUAD), lung squamous cell carcinoma (LUSC), prostate adenocarcinoma (PRAD), bladder urothelial carcinoma (BLCA), and thyroid carcinoma (THCA).

In addition, CD93 serves as a stable prognostic biomarker across most cancer types, with the exception of ACC, BRCA, CHOL, diffuse large B-cell lymphoma (DLBC), PAAD, PRAD, SKCM, and UCS. Single-cell sequencing analysis indicates that macrophages, astrocytes, cancer-associated fibroblasts (CAFs), T cells, B cells, ECs, neutrophils, and cancer cells are the primary CD93-expressing cells within the TME. Furthermore, in multiple cancer types, CD93 expression positively correlates with the expression of various immune checkpoint molecules, including CD86, leukocyte-associated immunoglobulin-like receptor 1 (LAIR1), V-set immunoregulatory receptor (VSIR), and neuropilin 1 (NRP1). Biomarker correlation analysis across 25 immunotherapy cohorts revealed that in 8 cohorts, CD93 had the same predictive value as tumor mutation burden (TMB), a finding corroborated by a pan-cancer analysis. Gene set enrichment analysis (GSEA) in COAD, BLCA, KIRC, and LIHC showed that the high CD93 expression group was enriched in gene sets related to cell migration and tissue migration pathways, which are closely associated with tumor angiogenesis. Additionally, pan-cancer survival analysis reveals that elevated CD93 expression generally poses a risk in most tumors, correlating with advanced TNM stages. In contrast, in KIRC, higher CD93 levels are linked to better prognosis and lower TNM stages, suggesting a protective role. The study also highlights a significant relationship between CD93 expression and immune infiltration, showing strong correlations with immunomodulators and immunotherapy markers such as TMB and MSI. Analysis using the TIMER database identified high CD93 expression in THCA and significant downregulation in HNSC, contradicting Guo et al.’s findings. Zhang et al. synthesized data from various sources, largely supporting prior analyses, and noted that CD93 overexpression is associated with poor prognosis in cancers like LGG, ovarian serous cystadenocarcinoma (OV), KIRP, and LUSC. The positive correlation between CD93 and M2 macrophages suggests an immunosuppressive role in the tumor microenvironment. Additionally, high CD93 expression is primarily linked to inflammation and angiogenesis, underscoring its critical role in cancer, inflammatory diseases, and homeostasis [161].

Subsequently, the published literature has extensively explored the role of specific genes in various cancer types using public databases. In 2023, Jiang et al. reported that both the transcriptional and protein expression levels of CD93 were significantly elevated in hepatocellular carcinoma, and were closely associated with patient prognosis. Furthermore, the function of CD93 in different cancers was primarily related to the IGFBP7/CD93 signaling pathway. Notably, CD93 expression was positively correlated with the infiltration levels of several immune cell types, including B cells, CD8^+^ T cells, CD4^+^ T cells, macrophages, neutrophils, and dendritic cells [162]. From 2023 to 2025, multiple research teams consistently identified CD93 as a gene significantly associated with gastric cancer (GC). Specifically, Wu, Shen et al. reported that in stomach adenocarcinoma, CD93 was predominantly expressed on vascular ECs, and its expression level in cancer tissues was significantly higher than in adjacent normal gastric tissues. CD93 was linked to poor prognosis of STAD, the abundance of multiple immune cell infiltration levels, and neutrophil extracellular traps (NETs), which are crucial for cancer occurrence and metastasis, and could also serve as a prognostic indicator for gastric cancer patients. In STAD, CD93 expression inversely correlated with CD93 mutations and methylation, while positively associating with most immunosuppressive genes (such as PD-1, PD-L1, CTLA-4, and LAG3), immunostimulatory genes, HLA, chemokines, and chemokine receptors. Zhu et al. identified CD93 as an independent prognostic marker for gastric cancer survival and a predictor of chemotherapy sensitivity [163,164,165,166,167]. Similarly, CD93 was part of the NET-related gene set in bladder cancer [168]. Qu et al. found elevated CD93 expression in lung squamous cell carcinoma, associating it with cancer-related pathways. CD93 was shown to contribute to chemotherapy and immunotherapy resistance by enhancing tumor cell stemness, reducing immune cell infiltration, and inducing T-cell exhaustion. Patients with low CD93 expression exhibited higher chemotherapy and immunotherapy response rates, while high expression was associated with advanced clinical stages and poor prognosis [169]. Barrett’s esophagus (BE) is a well-established risk factor for esophageal adenocarcinoma (EAC). Previous studies have confirmed the expression of gastric H^+^/K^+^ ATPase proton pump and pepsin in some cases of BE. Stabenau et al. reported that the primary upstream regulators of non-coding transcripts differentially expressed in the human Becell line BAR-T, which expresses pepsinogen and/or proton pump included CD93, p300-CBP, and I-BET-151, all of which had been previously associated with EAC or carcinogenesis [170]. Furthermore, recent preclinical studies have suggested that the intake of non-steroidal anti-inflammatory drugs (NSAIDs), such as aspirin and naproxen, may be an effective intervention strategy for TMPRSS2-ERG fusion-driven prostate cancer. Prasad et al.’s analysis of plasma and prostate tissue samples identified 54 molecules with significantly altered expression in mice with prostate cancer revealing that NSAIDs’protective effects linked to reduced CD93 expression [171]. Related research has also identified CD93 as a characteristic gene of leukemia stem cells (LSCs) associated with acute myeloid leukemia (AML) recurrence [172].

Multiple studies have examined biopsy samples from cancer patients to explore the role of CD93 in tumor progression. Bao et al. reported elevated CD93 expression in nasopharyngeal carcinoma (NPC), which was linked to advanced T-stage, N-stage, distant metastasis, clinical stage, and poor prognosis [173]. Olsen et al. analyzed 101 colorectal cancer samples using various techniques, including ELISA, immunohistochemistry, Western blotting, gene expression analysis, and real-time PCR. They observed upregulated CD93 in tumor tissues compared to normal tissues and downregulated soluble CD93 in patient plasma compared to healthy individuals. Additionally, the rs2749817 T/T genotype was prevalent in stage IV colorectal cancer (CRC) patients and associated with increased mortality and recurrence risk [2,7]. Yang et al. corroborated these findings in a study of 134 CRC patients, demonstrating higher CD93 levels in CRC blood vessels compared to adjacent normal tissues and a positive correlation between CD93 protein expression and macrophage infiltration in CRC. CD93 expression was associated with tumor location and microsatellite instability, and high CD93 levels were linked to improved overall survival [174]. Furthermore, Langenkamp et al. assessed vascular CD93 expression in lesion samples from 235 high-grade astrocytoma patients using immunohistochemistry and found that higher CD93 expression in blood vessels was associated with poorer survival, with high-grade gliomas exhibiting a greater prevalence of CD93 positivity compared to low-grade gliomas [48]. Ma et al. reported a positive correlation between CD93 expression and the infiltration of immunosuppressive cells, such as macrophages, regulatory T cells, and myeloid-derived suppressor cells, in human gliomas [175]. Analyses of squamous cell carcinoma biopsies from lung cancer patients and ductal cancer cells from postmenopausal breast cancer patients revealed downregulated CD93 expression compared to controls. Furthermore, a SNP analysis in Korean women identified a cytosine-to-thymine polymorphism upstream of CD93, along with other innate immunity-related SNPs, as associated with increased breast cancer susceptibility [176]. Gastric cancer peritoneal metastasis (GCPM) significantly contributes to cancer-related mortality globally. Dong et al. conducted high-throughput RNA sequencing on clinical specimens, including primary non-metastatic gastric cancer, primary gastric cancer with peritoneal metastasis, and adjacent normal gastric mucosa, and identified CD93 as a key mRNA linked to gastric cancer peritoneal metastasis, which significantly contributes to cancer-related mortality globally [177]. Another study analyzing RNA-seq data from 22 patients with esophagogastric junction adenocarcinoma(AEG) suggests that CD93 functions as an oncogene with its expression linked to T-stage and maximum tumor diameter [178]. These results imply that CD93 may exert an immunosuppressive effect within the tumor microenvironment.

How does CD93 exert its immunosuppressive effects within the tumor microenvironment? Several studies have explored this. Glioblastoma, a highly aggressive astrocytic brain tumor, is characterized by microvascular proliferation and abnormal vasculature. In 2012, Dieterich et al. identified 95 genes differentially expressed in glioblastoma blood vessels, highlighting increased CD93 gene and protein expression, which correlated with tumor grade, reaching its highest in grade IV gliomas [179]. In 2015, Langenkamp et al. analyzed microarrays from 235 human control brain biopsies and glioma tissues of grades II, III, and IV. They found CD93 positivity in the blood vessels of 73% of grade IV glioma samples, particularly in the tumor core and invasive front, with vascular CD93 expression correlating with glioma grade. In vitro CD93 knockdown inhibited tubular structure formation, endothelial cell adhesion and migration. Using a GL261 glioma mouse model, they showed CD93 was predominantly expressed in glioma vasculature, with stronger staining in tumor blood vessels than in surrounding brain tissue. In female mice, CD93 deletion delayed GL261 glioma growth and enhanced survival compared to wild-type mice, linked to increased permeability and reduced perfusion of glioma vasculature. CD93 silencing in ECs resulted in the displacement and loss of VE-cadherin at intercellular junctions, increased stress fiber formation and cytoskeletal rearrangement reducing cell–matrix adhesion and inhibiting cell spreading. These findings suggest that targeting CD93 with antibodies could inhibit tumor angiogenesis. Notably, leukocyte infiltration showed no differences between CD93-deficient and wild-type mice in GL261 gliomas and T241 fibrosarcomas. Interestingly, CD93 deletion delayed tumor growth only in female mice with GL261 gliomas and T241 fibrosarcomas, but impaired glioma vasculature perfusion equally in both sexes [48]. In 2018, Lugano et al. confirmed CD93 upregulation in glioblastoma blood vessels compared to normal brain vessels and identified a CD93/MMRN2/β1 integrin/fibronectin signaling axis, revealing a new mechanism for integrin activity regulation in cancer angiogenesis. They demonstrated that CD93 was expressed in endothelial filopodia and could promote filopodia formation.CD93 interacts with MMRN2 to stabilize its presence in endothelial filopodia. The CD93-MMRN2 complex is essential for β1 integrin activation, FAK phosphorylation, and fibronectin fibril formation in ECs. In gliomas implanted in CD93-deficient mice, β1 integrin activation in tumor vasculature was impaired, preventing fibronectin from forming fibrous structures [56]. Bao demonstrated CD93’s critical role in nasopharyngeal carcinoma progression and angiogenesis, showing that silencing CD93 in the CNE2 cell line significantly reduced proliferation [173]. In breast cancer, Liu et al. reported high CD93 expression in pathological samples and cell lines, highlighting its role in promoting MDA-MB-231 cell proliferation, migration, and VM. Silencing CD93 and MMRN2 inhibited integrin β1 activation, suppressing the PI3K/AKT/SP2 signaling pathway and reducing breast cancer growth and VM [77].

The recent proposal of a novel perspective has intensified the focus on the mechanisms underlying tumor-related angiogenesis. Pathological angiogenesis has long been recognized as a hallmark of malignancy [180]. Aberrant tumor blood vessels are often characterized by leakage, disorganization, and functional anomalies, contributing to a metabolic microenvironment marked by acidosis, hypoxia, and disrupted glucose metabolism [181]. These conditions support tumor persistence, facilitate immune evasion, and promote tumor growth and metastasis, thereby complicating immunotherapy [182]. The goal of normalizing tumor vasculature is to establish a functional vascular network, alleviate hypoxia, enhance vascular perfusion [183,184], and improve immune cell infiltration into tumors [185]. In 2021, Sun et al. identified an interaction between CD93 and IGFBP7, implicating this pathway in vascular abnormalities [63]. In CD93^−/−^ mouse models, tumor vasculature showed increased permeability and reduced perfusion, suggesting that vascular dysfunction might partly underlie the anti-tumor effects of CD93 deficiency [48,56]. Conversely, treatment with a CD93 monoclonal antibody promoted vascular maturation and reduced leakage, thereby alleviating hypoxia and enhancing perfusion. This discrepancy may arise from the complete absence of CD93 in CD93^−/−^ mice, whereas antibody-mediated blockade of CD93 signaling is transient and partial. Additionally, blocking the CD93 signaling pathway enhances the presence of effector T cells within tumors, thereby increasing their sensitivity to immune checkpoint therapy. This research suggests that inhibiting CD93 primarily improves vascular function, and targeting the IGFBP7/CD93 pathway could be a safe, durable strategy to enhance tumor vasculature [63]. Vemuri et al. further assessed CD93’s role in vascular integrity in metastatic cancers, finding it crucial for maintaining the endothelial barrier and limiting metastasis. In CD93 knockout (CD93^−/−^) mice, primary melanoma growth is inhibited, but metastasis increases due to disrupted adherens and tight junctions in tumor ECs and elevated matrix metalloproteinase 9 expression at metastatic sites. CD93 interacts with VEGFR2, and its absence results in VEGF-induced overphosphorylation of VEGFR2 in ECs. CD93 deficiency destabilizes blood vessels in primary tumors, facilitates tumor cell invasion into the vasculature, and promotes a conducive microenvironment at metastatic sites. This result contradicts the findings of Sun et al. Vemuri et al. suggest that administering an anti-CD93 neutralizing antibody to tumor-bearing mice, which disrupts the CD93 and IGFBP7 interaction, is associated with vascular normalization. Unlike complete gene deletion in CD93^−/−^ mice, this antibody does not inhibit all CD93 functions, as CD93 interacts with various molecular partners [186]. In 2024, research by Jiang, Zhang, and Pan established robust association between CD93 and immune cell infiltration in tumors. Jiang et al. found that monocytes in the peritumoral tissue of hepatocellular carcinoma (HCC) showed markedly increased CD93 expression. These CD93-positive monocytes were positively correlated with peritumoral CD8^+^ T cell density and inversely correlated with intratumoral CD8^+^ T cell density. In vitro studies indicated that tumor-induced monocytes upregulate CD93 via a glycolytic switch through the extracellular Erk pathway. CD93 promotes programmed death-ligand 1 (PD-L1) expression through the protein kinase B (PKB)-glycogen synthase kinase 3β (GSK3β) axis and stimulates monocytes to produce versican, which impedes CD8^+^ T cell migration by interacting with hyaluronic acid and collagen. Targeting CD93 on monocytes enhanced CD8^+^ T cell infiltration and activation, increasing tumor susceptibility to anti-PD-1 therapy in mice. This highlights CD93-positive monocytes as a promising immunotherapeutic target in HCC treatment [187]. The study by Zhang et al. elucidated the pivotal role of CD93 in modulating anti-lung tumor immunity. The authors proposed that CD93 expressed on pleural mesothelial cells inhibits systemic anti-lung tumor T cell responses by suppressing CCL21 secretion and reducing dendritic cell migration to lung tumors [188]. Furthermore, Pan et al. investigated the interplay between immune cells, inflammatory factors, and cancer, identifying five immune cell types as potential mediators. Notably, CD4^+^ on activated regulatory T cells expressing CD39 were found to mediate 12.384% of the increased cholangiocarcinoma risk associated with CD93 [189]. Building upon this, Sun et al. suggested in 2025 that blocking CD93 could enhance intratumoral T cell infiltration by upregulating adhesion molecules (intercellular cell adhesion molecule-1 and vascular cell adhesion molecule-1) on tumor blood vessels, with T cells predominantly infiltrating tumors characterized by dysfunctional or immature vasculature [190].

Conventional cancer treatments have traditionally relied on surgery, radiotherapy, and chemotherapy. However, recent decades have witnessed the emergence of innovative immunotherapies, which offer effective treatment with minimal side effects across various cancer types. This approach harnesses the immune system to target cancer cells, utilizing compounds such as immune checkpoint inhibitors (ICIs) that modulate pathways like PD-1/PD-L1 and CTLA-4, as well as cytokines, chimeric antigen receptor (CAR)-T cells, and monoclonal antibodies [191]. Despite the purported breakthroughs in cancer immunotherapy over the past decade, many patients have not experienced clinical benefits [192].

The study by Sun et al. in 2021 demonstrated that combining immunotherapy with a CD93 blocker enhances tumor sensitivity to immune checkpoint blocker therapy, improving treatment response in preclinical tumor models [63]. The use of a CD93 monoclonal antibody was shown to promote vascular maturation, alleviate tumor hypoxia, and increase tumor perfusion without significantly affecting vascular homeostasis in healthy organs, unlike anti-VEGFR monoclonal antibodies. This approach not only facilitates chemotherapy drug delivery, enhances the anti-tumor response to gemcitabine or fluorouracil, and significantly increases intratumoral effector T cells when the CD93 pathway is inhibited, rendering mouse tumors more responsive to immune checkpoint treatment. In B16 melanoma and KPC pancreatic cancer mouse models, the addition of an anti-CD93 antibody enhanced the anti-tumor effects mediated by immune checkpoint blockers, such as PD-1 and CTLA-4 monoclonal antibodies, inhibiting tumor growth and prolonging mouse survival [63]. Furthermore, the study by Zhang et al. in 2024 showed that the CD93 blocking antibody (anti-CD93) surpasses the VEGFR blocking antibody in suppressing lung tumor growth [188]. Anti-CD93 not only inhibits tumor angiogenesis but also enhances CCL21 secretion by pleural mesothelial cells (pMCs), making it more effective than VEGF receptor blockers. Additionally, anti-CD93 can overcome lung tumor resistance to anti-PD-1 therapy, offering a promising strategy for lung cancer treatment [188]. Sun’s recent study demonstrates that in tumors with an upregulated CD93 pathway, anti-CD93 selectively increases effector T cell infiltration, enhancing CAR-T cell infiltration and function in solid tumors. In mouse models, blocking the CD93 pathway significantly improves CAR-T cell therapy efficacy and synergizes with adoptive cell therapy (ACT) to promote tumor vascular maturation [190].

The upregulation of CD93 on tumor-associated ECs and the promising results from blocking CD93 signaling have catalyzed numerous preclinical and clinical trials. Miao et al. identified two novel nanobodies, NC81 and NC89, from a shark-derived phage library, expressed them in E. coli, and achieved a purity exceeding 95%. Both nanobodies significantly inhibited angiogenesis and increased vascular permeability in vitro [193]. Liang et al. developed a nanodroplet-based ultrasound contrast and therapeutic agent (NDsUCA/Tx) that targets tumor vasculature with high CD93 expression, enabling molecular imaging of tumor neovascularization and precise tumor microenvironment remodeling [194]. Furthermore, Wang et al. created CD93-targeted microbubbles (MBs) with a gaseous core and a hybrid membrane, incorporating MMRN2 and lipid. These microbubbles demonstrated effective CD93 recognition and accumulation in CD93-rich tumor areas both in vitro and in vivo. Contrast-enhanced ultrasound (CEUS) imaging with these microbubbles showed a negative correlation between targeted ultrasound intensity and the inflammatory tumor immune microenvironment, as well as cytotoxic T cell infiltration indicating that elevated CD93 expression in tumor vasculature correlates with a poor immune response in prostate cancer, positioning CD93 on ECs as a distinct marker of an immunosuppressive microenvironment [195]. Richards et al. demonstrated that CD93 CAR T cells, engineered with a novel humanized CD93-specific binder, effectively target and eliminate AML cells both in vitro and in vivo, while sparing hematopoietic stem and progenitor cells (HSPCs) [196]. Clinically, lenalidomide, an approved oral myeloma treatment, was reformulated into a nanosuspension-based hydrogel for transdermal application as a percutaneous neoadjuvant therapy. Zhang et al. confirmed lenalidomide’s ability to reverse PI3K-AKT pathway dysregulation and CD93 overexpression, while promoting the infiltration of CD8^+^ T cells, CD4^+^ T cells, and dendritic cells into the tumor. Preoperative administration of Len-NBHs was shown to inhibit melanoma spread, reduce tumor size, and consequently decrease the surgical resection area, thereby enhancing patient survival and prognosis [197].

Multiple experiments abroad have been conducted to develop CD93-related drugs, including DCBY 02, WO2024115637, CD34-Derived Allogeneic Dendritic Cell Cancer Vaccine, and DCSZ11. Concurrently, several research institutions in China have made similar efforts, such as SY2053, Anti CD93 antibody (China Resources Biopharmaceutical), CN116271112, and JS013 (as shown in Table 2). DCBY 02, Anti CD93 antibody (China Resources Biopharmaceutical), JS013, and DCSZ11 are all CD93 monoclonal antibodies aimed at treating cancer by inhibiting CD93. The development of DCBY 02 has been terminated, and there has been no progress in the development of JS013. The clinical development of Anti CD93 antibody (China Resources Biopharmaceutical) remains in the preclinical stage. However, DCSZ11 entered Phase 2 clinical trials (NCT07035249) on July 15, 2025, targeting metastatic solid tumors and head and neck squamous cell carcinoma. CN116271112 (Patent (CN116271112A)) is an antibody-conjugated radionuclide drug, and WO2024115637 (Patent (WO2024115637A1)) is a CD93 small molecule inhibitor, both of which entered the drug discovery stage in 2023 and 2024, respectively. CD34-Derived Allogeneic Dendritic Cell Cancer Vaccine is a therapeutic vaccine that can treat not only tumors but also diseases related to the digestive system, endocrine, and metabolism, and it entered the preclinical stage on 28 April 2025, with a focus on pancreatic cancer (data are referenced from https://synapse.zhihuiya.com/).

### 6.5. Other Diseases

CD93 plays a crucial role in endothelial homeostasis and angiogenesis, prompting investigation into its impact on diseases associated with pathological vasculature. In 2023, Piani et al. examined early pregnancy serum CD93 levels in women with recent pre-eclampsia (PE) compared to those with normotensive pregnancies (NP). They found significantly lower serum CD93 levels in women with PE than in those with NP (*p* < 0.001), indicating a negative correlation between serum CD93 levels and PE risk. This suggests that CD93 may influence placentation, potentially causing angiogenesis defects, vascular dysfunction, and the onset of PE [198].

Age-related macular degeneration (AMD) is a leading cause of vision loss among the elderly in developed nations. Late-stage AMD is driven by excessive neovascularization of choroidal capillaries, leading to vascular leakage, hemorrhage, and fibrosis [199,200]. Despite inhibiting VEGFR-A, neovascularization persists, underscoring the need for new endothelial cell targets. In 2017, Tosi et al. reported overexpression of both transmembrane and sCD93 in patients with neovascular AMD. Using immunofluorescence, they detected significant CD93 staining in von Willebrand factor-positive ECs within choroidal neovascular membranes. Similar patterns were observed in intra- and extraocular tumor vessels, whereas normal choroidal vessels exhibited weak CD93 staining. Furthermore, soluble CD93 levels in the aqueous humor of newly diagnosed neovascular AMD patients were significantly elevated, but decreased, albeit not significantly, following anti-angiogenic treatment [201]. Three years later, Tosi and colleagues observed that hyperproliferative choroidal ECs in AMD patients exhibited elevated CD93 levels compared to healthy controls. The CD93-MMRN2 interaction and CD93 knockout mice exhibited significantly reduced neovascularization, with choroidal ECs demonstrating diminished sprouting ability in in vitro angiogenesis assays. This suggests that CD93 and its interaction with multimerin-2 are crucial in choroidal pathological angiogenesis, presenting potential therapeutic avenues for neovascular AMD [202]. Recent research by Raucci et al.’s recent research corroborates these findings, demonstrating that 4E1, a murine monoclonal antibody against CD93 [54], and its derivative sc-4E, effectively inhibit endothelial cell proliferation, migration, sprouting, and capillary-like structure formation, highlighting their promise as treatments for ocular vascular diseases driven by pathological angiogenesis [203].

The field of neurological and psychiatric diseases has identified CD93 as one of the four proteins significantly associated with white matter lesion volume among the 28 proteins enriched in brain endothelial cells (BECs), the primary cell type of the blood–brain barrier. BECs may serve as sensitive plasma biomarkers for cerebral small vessel disease and dementia [204]. Liang et al. found that CD93 plays a crucial negative regulatory role in astrocyte genesis in the mouse cerebral cortex. Knockout of the CD93 gene in the late embryonic stage led to decreased neuron production and increased astrocyte production, as CD93 interacts with MMRN2 to inhibit astrocyte genesis. Furthermore, CD93 knockout mice exhibited autism-like behaviors, providing new insights into the pathogenesis of psychiatric diseases [205]. Additionally, Wang et al. used public databases to identify CD93 as one of the hub genes related to the complement system in Parkinson’s disease (PD). Through immune infiltration analysis, they found that these hub genes are significantly correlated with various immune cells, such as myeloid-derived suppressor cells and macrophages, offering a new perspective for the diagnosis and treatment of PD [206].

Interestingly, the expression of CD93 has been associated with active tuberculosis (TB) and polycystic ovary syndrome (PCOS). Smoking is a well-established risk factor for TB, doubling the risk. Piani et al. reported an enrichment of immature inflammatory monocytes with a phenotype indicative of recent recruitment, continuous differentiation, enhanced activation, and a state akin to chronic obstructive pulmonary disease in the lungs of smokers compared to non-smokers. Using integrated single-cell RNA sequencing and flow cytometry, the researchers identified CD93 as a marker for these smoking-related lung monocyte subsets [207]. Additionally, a bioinformatics and Mendelian randomization (MR) study found a significant association between 10 key genes, including CD93, and the risk of PCOS [208]. Martínez-Moro et al. in 2023 investigated the potential association between CD93 expression and blastocyst development [209]. Transcriptomic analysis of cumulus cells revealed that CD93 expression was downregulated in blastocysts that reached the blastocyst stage but did not result in successful pregnancy (P^−^) and those that reached the blastocyst stage and led to a clinical pregnancy (P^+^) compared to those that failed to develop to the blastocyst stage(Bl^−^) [209] (see Figure 3 for the associations between CD93 and various diseases).

## 7. Conclusions

This review elaborates on the structural characteristics of CD93 and related family members, noting their structural similarities and shared ligand-binding capabilities. For example, both CLEC14A and CD93 bind to MMRN2, while CLEC14A also interacts with heat shock protein 70-1 (HSP70-1), a ligand shared with thrombomodulin [210]. These proteins thus have overlapping cellular functions in both physiological and pathological contexts. We emphasize that CD93 is expressed in various cell types, with high expression on ECs, monocytes, neutrophils, platelets, and microglia, where it mediates key physiological processes. CD93 interacts with intracellular proteins like moesin, GIPC1, and Cbl to regulate angiogenesis, cell adhesion, and migration. It also engages with extracellular ligands such as MMRN2 and IGFBP7, playing a critical role in maintaining tumor blood vessel stability and regulating permeability.

In the context of clinical diseases, CD93 has been implicated in a range of pathological conditions, including inflammation, cardiovascular diseases, autoimmune disorders, and various cancers (as shown in Figure 3). Notably, both soluble and membrane-bound forms of CD93 are significantly elevated in several inflammatory conditions, such as asthma, peritonitis, and periodontitis. Soluble CD93 has been shown to activate monocytes and promote the secretion of inflammatory cytokines. Autoimmune diseases are also closely associated with inflammatory responses, as evidenced by the significant correlation between CD93 expression and conditions like systemic lupus erythematosus and systemic sclerosis. CVD poses a significant global health threat and economic burden. Accumulating evidence suggests that CD93 may play a crucial role in various cardiovascular conditions, with differential expression observed across multiple CVDs, including diabetes, coronary heart disease, and atherosclerosis. These findings indicate that CD93 could serve as a novel biomarker or therapeutic target for cardiovascular risk assessment, prevention, and treatment; however, further research is required to elucidate its precise clinical value. CD93, integral to endothelial homeostasis and angiogenesis, has also been implicated in pathological conditions characterized by abnormal angiogenesis. Altered CD93 expression has been detected in the choroidal ECs of patients with age-related macular degeneration and pre-eclampsia. Experimental studies demonstrate that antibodies or compounds inhibiting CD93 expression can suppress neovascularization in these conditions, underscoring its potential as a therapeutic target.

The role of CD93 in oncology has garnered increasing attention in recent years. While current anti-angiogenic immunotherapies primarily aim to disrupt the tumor vasculature and limit nutrient delivery, emerging evidence suggests that vascular normalization may be a more sustainable and effective strategy for targeting tumor angiogenesis [13]. Multiple studies have demonstrated that CD93 is significantly overexpressed in tumor-associated vasculature and is associated with tumor malignancy, progression, metastasis, and immune infiltration. Furthermore, reduced CD93 expression has been correlated with improved prognosis and enhanced response to immunotherapy in cancer patients. This suggests that CD93 may play an immunosuppressive role in the tumor microenvironment, and that targeting the CD93-related signaling pathway may represent a novel therapeutic strategy for tumor immunotherapy. Preclinical studies have shown that CD93 monoclonal antibodies, whether administered as monotherapy or in combination with CAR T cell therapy, immune checkpoint inhibitors, or chemotherapeutic agents in murine models, exhibit significant antitumor activity. Therefore, CD93-targeted therapies are expected to receive increasing attention in the coming years. The development of humanized monoclonal antibodies or other therapeutics targeting CD93 may offer new treatment options for patients with inflammatory disorders, cardiovascular diseases, and various cancers, potentially leading to improved clinical outcomes. Currently, multiple clinical trials investigating CD93-directed interventions are underway, with the DCSZ-11 compound recently progressing to Phase 2 evaluation. The continued exploration of this molecular target holds significant potential for future clinical applications.

## Figures and Tables

**Figure 1 ijms-26-08617-f001:**
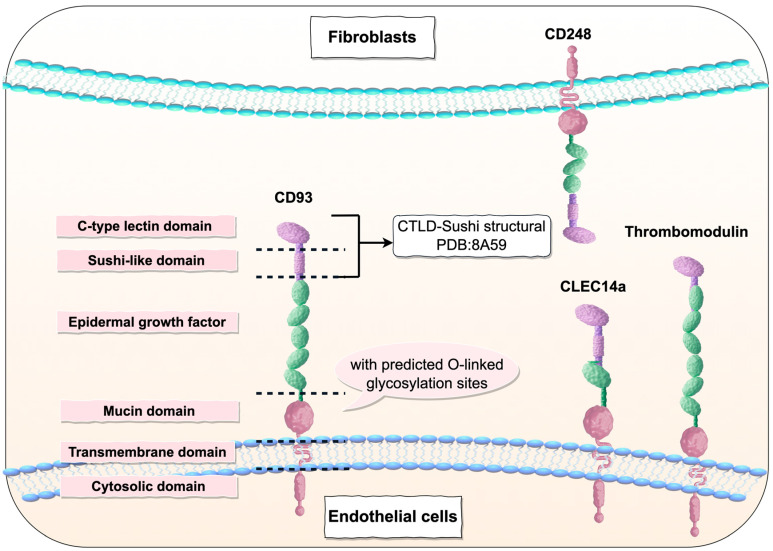
Structural representation of group XIV C-type lectin domain proteins. CD93 features an N-terminal signal peptide, a C-type lectin-like domain (CTLD) with conserved cysteine residues, a sushi-like domain (not present in Thrombomodulin), five epidermal growth factor (EGF)-like domains (compared to six in Thrombomodulin, three in CD248, and one in CLEC14a), a mucin-like region with predicted O-linked glycosylation sites, a transmembrane region, and a cytoplasmic tail. And the structure of the CTLD and sushi domain of CD93 (CTLD-Sushi structural) has been confirmed by PDB deposition under the accession code 8A59.

**Figure 2 ijms-26-08617-f002:**
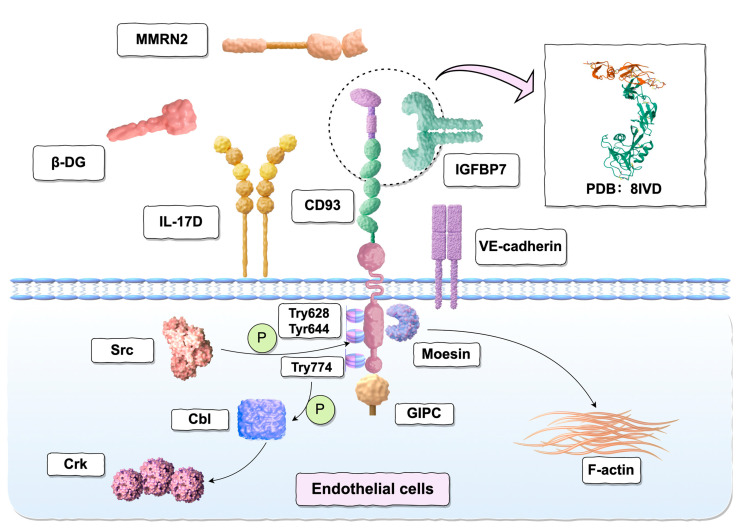
The various binding ligands and interacting proteins of CD93. The extracellular domain of CD93 can bind to several ligands, including multimerin-2 (MMRN2), insulin-like growth factor binding protein 7 (IGFBP7), β-dystroglycan (DG), interleukin-17D (IL-17D) and vascular endothelial cadherin (VE- cadherin). The region where CD93 and IGFBP7 bind has been confirmed by PDB deposition under the accession code 8IVD (this structure diagram is referenced from Protein Data Bank). The intracellular domain of CD93 can interact with the adaptor protein Cbl, moesin and GIPC. Binding of CD93 to β-DG leads to phosphorylation of tyrosine residues Tyr628 and Tyr644 in the cytoplasmic tail of CD93 by Src kinase, which stimulates the phosphorylation of the downstream effector Cbl. Phosphorylation of Cbl at Tyr774 may enable it to function as an adaptor protein, facilitating the recruitment of Crk and inducing cell migration and tubular structure formation. Additionally, the interaction between CD93 and GIPC can mediate polarized transport through the branching of linear actin fiber bundles.

**Figure 3 ijms-26-08617-f003:**
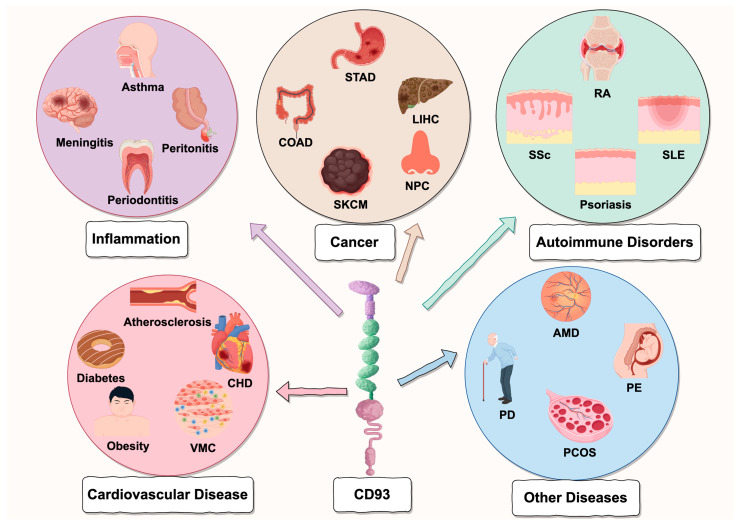
CD93 dysregulation in various diseases. In the context of clinical diseases, CD93 has been implicated in a range of pathological conditions, including inflammation (e.g., asthma, peritonitis, meningitis, periodontitis, etc.), cardiovascular diseases (e.g., atherosclerosis, diabetes, obesity, viral myocarditis (VMC), coronary heart disease (CHD), etc.), autoimmune disorders (e.g., rheumatoid arthritis (RA), systemic sclerosis (SSc), systemic lupus erythematosus (SLE), psoriasis, etc.), various cancers (e.g., stomach adenocarcinoma (STAD), colon adenocarcinoma (COAD), skin cutaneous melanoma (SKCM), nasopharyngeal carcinoma (NPC), liver hepatocellular carcinoma (LIHC), etc.), and other diseases (e.g., age-related macular degeneration (AMD), pre-eclampsia (PE), Parkinson’s disease (PD), polycystic ovary syndrome (PCOS), etc.).

**Table 1 ijms-26-08617-t001:** Summary of group XIV CTLDs’ characteristics.

	CD93	TM	CD248	CLEC14a
Alternative name	C1qR1, C1qRp, AA4	CD141, BDCA3	Endosialin, TEM1	/
Structural composition	CTLD, sushi-like domain, five EGF-like domains, mucin domain, transmembrane domain, cytosolic domain	CTLD, six EGF-like domains, mucin domain, transmembrane domain, cytosolic domain	CTLD, sushi-like domain, three EGF-like domains, mucin domain, transmembrane domain, cytosolic domain	CTLD, sushi-like domain, one EGF-like domains, mucin domain, transmembrane domain, cytosolic domain
Mainly represent cell types	ECs, neurons, and various myeloid cells (such as macrophages, monocytes, and stem cells)	Vascular endothelial cells, lymphatic endothelial cells, mesothelial cells, astrocytes, keratinocytes, osteoblasts, chondrocytes, alveolar epithelial cells, and various hematopoietic cells	Fibroblasts, smooth muscle cells, pericytes, mesenchymal stem cells, and naive T cells	ECs
Normal adult tissues	Extremely low expression or complete absence	Moderate to high expression	Absence (e.g., in the brain, stomach, skin, ovaries) or low expression	Absent or minimally expressed
Combining ligands and interacting proteins	MMRN2, β-dystroglycan, Cbl, IGFBP7, Moesin, GIPC, VE-cadherin, IL-17D	Thrombin, protein C, fibronectin, Lewis Y antigen, GPCR15, ezrin	Extracellular matrix proteins, including collagen type I, collagen type IV, and fibronectin	MMRN2 and HSP70-1A
Physiological function	Clearance of apoptotic cells, endothelial cell maturation and migration, intercellular adhesion, promotion of angiogenesis	Promote embryonic development, mediate cell adhesion and inflammation regulation, anticoagulate, regulate angiogenesis	Reshape the developing vascular system, mediate the migration, activation and proliferation of stromal cells, promote inflammation	Endothelial cell migration and adhesion, regulation of lumen formation
Association with inflammatory diseases	It is upregulated in various inflammatory diseases, such as asthma and peritonitis.	Exerts anti—inflammatory effects by inactivating pro—inflammatory cytokines and inhibiting the inflammatory complement proteins C3a and C5a.	Promote inflammation and upregulate in fibroblasts and pericytes of synovial tissue as well as mesenchymal cells of the skin.	Undefined
Relationship with tumors	It is highly expressed in various cancers, and its expression level is positively correlated with tumor malignancy and poor prognosis, and negatively correlated with survival rate.	In various cancers, high expression of membrane-bound TM is negatively correlated with tumor aggressiveness and positively correlated with survival rate; in contrast, soluble TM shows the opposite pattern.	High expression levels are positively correlated with tumor invasion and metastasis and negatively correlated with survival rate.	Highly expressed in various cancers.
Preclinical/preclinical therapies	Monoclonal antibodies, therapeutic vaccines, small-molecule chemical drugs, antibody-conjugated radionuclides, dendritic cell vaccines	Monoclonal antibodies, protein-based drugs	Antibody-conjugated radionuclides, small-molecule chemical drugs, Monoclonal antibodies, therapeutic radiopharmaceuticals, bispecific antibodies, CAR T therapy, therapeutic vaccines, mRNA vaccines	Monoclonal antibodies, CAR T therapy
Clinical trial status	1. Drug discovery (WO2024115637, CN116271112) 2. Preclinical (SY2053, CD34-Derived Allogeneic Dendritic Cell Cancer Vaccine, Anti CD93 antibody) 3. Phase 1 clinical trial (DCSZ-11) 4. Termination (DCBY020) 5. No progression (JS013)	1. Drug discovery (CN114686444, CN118599002, CN114686443) 2. Preclinical (AD-010)	1. Preclinical ([Ac-225]-FPI-1848, ANP-021, [177 Lu]-FPI-1835, Endosialin-directed E3K CAR-T cells) 2. Termination (Ontuxizumab, MP-ENDOS-1959, AVS300)	1. Preclinical (Anti-clec14a antibodies) 2. No progression (Anti CLEC14a-CTLD antibody, CLEC14A-specific CAR T cells)

TM thrombomodulin, C1qRp complement component C1q receptor, CTLD C-type lectin-like domain, ECs endothelial cells, EGF epidermal growth factor, MMRN2 multimerin-2, IGFBP7 insulin-like growth factor-binding protein 7, VE-cadherin vascular endothelial cadherin, IL-17D interleukin-17D, GPCR15 G protein-coupled receptor 15, CAR chimeric antigen receptor.

**Table 2 ijms-26-08617-t002:** Summary of investigational drugs related to CD93.

Drug Name	Mechanism of Action	Drug Type	Indications Under Research	Original Research Institution	Research Institutions Under Development	The Highest Research and Development Stage
DCSZ-11 (DCSZ11)	CD93 inhibitor	Monoclonal antibodies	Advanced malignant solid tumors	DynamiCure Biotechnology LLC	DynamiCure Biotechnology LLC	Phase 2 clinical trial (2025-07-15)
SY2053	CD93 inhibitor	Antibodies	Tumors	Shanghai Symray Biopharma Co., Ltd.	Shanghai Symray Biopharma Co., Ltd.	Preclinical
CD34-Derived Allogeneic Dendritic Cell Cancer Vaccine	CD40L regulator CD93 modulators CXCL13 modulators	Therapeutic vaccines, dendritic cell vaccines	Pancreatic cancer	Renovaro, Inc.	Renovaro, Inc.	Preclinical
Anti CD93 antibody (China Resources Biopharmaceutical)	CD93 inhibitor	Monoclonal antibodies	Tumors	China Resources Biopharmaceutical Co., Ltd.	China Resources Biopharmaceutical Co., Ltd.	Preclinical
WO2024115637	CD93 inhibitor	Small-molecule chemical drugs	Tumors	University of Bern	University of Bern	Drug discovery
CN116271112	/	Antibody-conjugated radionuclides	Tumors	The Fourth Military Medical University	The Fourth Military Medical University	Drug discovery
DCBY0 (DCBY 02, DCB7-02)	CD93 inhibitor	Monoclonal antibodies	/	DynamiCure Biotechnology LLC	/	Termination (Phase 1 clinical trial)
JS013	CD93 inhibitor	Monoclonal antibodies	/	Shanghai Junshi Biosciences Co., Ltd.	/	No progression (Preclinical)

Data are referenced from https://synapse.zhihuiya.com/.

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
