# Peer review of "CD93 in Health and Disease: Bridging Physiological Functions and Clinical Applications"

_ijms, 2025, doi:10.3390/ijms26178617_

Round 1

Reviewer 1 Report

Comments and Suggestions for Authors

1.Improve clarity and readability by refining sentence structure, ensuring concise expression, and maintaining consistent use of terminology throughout.

2.Optimise the visual presentation of tables and figures to maximise accessibility, incorporating clear annotations and succinct explanatory notes to highlight key messages.

3.Streamline the organisation of related content to reduce redundancy, strengthen logical transitions, and maintain a coherent narrative across sections.

4.Increase the proportion of references from the past five years to ensure the discussion reflects the most recent advances and maintains relevance to current research in the field.

Author Response

Dear Reviewer 1, 

Thank you very much for taking the time to review the manuscript. We have adopted your comments "The English could be improved to more clearly express the research." and have polished the manuscript sentence by sentence. For the convenience of discussion, we highlight the changes in red in the revised manuscript.

Thank you again for your valuable comments and suggestions. We look forward to 
hearing from you regarding our submission and we would be glad to respond to any 
further questions and comments that you may have. 

Yours sincerely, 
Menghan Cai

Reviewer 2 Report

Comments and Suggestions for Authors

The review article "CD93 in Health and Disease: Bridging Physiological Functions and Clinical Applications" by Cai et al. addresses a relevant topic of CD93, which plays roles in angiogenesis, inflammation, and immune regulation, has emerged as a promising therapeutic target in cancer, cardiovascular disease, and chronic inflammation. The authors provide a broad overview of CD93’s physiological and pathological roles very extensively, which I like a lot, but the description should be stronger also in the  structural part .

While the topic is timely and important, especially given the growing interest in targeting CD93 for vascular normalization and immunotherapy, the structural biology section is underdeveloped. The authors should incorporate experimentally resolved structures such as PDB: 8A59 and PDB: 8IVD, which reveal domain-level interactions and ligand-binding interfaces. A 2D ribbon diagram annotated with strand numbering, domain boundaries, and glycosylation sites would significantly enhance comprehension.

The figures throughout the manuscript are of poor quality. Figure 1 lacks adequate labeling and explanation, and Figure 2 appears to include elements from BioRender, yet the overall visual style is inconsistent and resemble more of PowerPoint graphics, which is confusing.

The statement that GST on CD93 allows it to bind to Moesin May be misleading and could confuse readers into thinking CD93 naturally contains a glutathione S-transferase domain. The section on GIPC is too brief and should be expanded to explain the functional consequences of CD93–GIPC interaction, such as its role in endocytosis and cytoskeletal dynamics. also, the IGFBP7 interaction deserves a more thorough molecular description, especially given the availability of structural data on pdb. The authors should detail the binding interface, the domains involved, and the implications for endothelial cell behavior and tumor vasculature.

The discussion of CD93-targeting drugs is currently in the conclusion, which undermines its importance. This content should be integrated into the main body of the review and expanded into a dedicated section. It should cover therapeutic strategies such as monoclonal antibodies and small molecules, and discuss their mechanisms, efficacy, and clinical development status in more details as it is definitely of interest to a broader audience.

Minor:

“Table 93”. line 56 page 2 i guess it should be “CD93”?

Author Response

Dear Reviewer 2,
Thank you very much for taking the time to review this manuscript and we have studied your comments point by point, revised the manuscript accordingly.
To facilitate this discussion, we first retype your comments in bold and then highlight the changes in red in the revised manuscript.
Comment 1:
The structural biology section is underdeveloped. The authors should incorporate experimentally resolved structures such as PDB: 8A59 and PDB: 8IVD, which reveal domain-level interactions and ligand-binding interfaces. A 2D ribbon diagram annotated with strand numbering, domain boundaries, and glycosylation sites would significantly enhance comprehension.
Respond: We are deeply appreciative of the constructive and insightful feedback provided by the reviewers. We regret any inconvenience this may have caused. Accordingly, we have supplemented the relevant PDB: 8A59 and PDB: 8IVD content in our manuscript and revised Figure 1(lines 108-109/lines 689-693).
“And the structure of the CTLD and sushi domain of CD93 (CTLD-Sushi structural) has been confirmed by PDB deposition under the accession code 8A59.”
“Sun et al. revealed that the interaction between CD93 and IGFBP7 is independent of IGF, relying instead on the CTLD structural domain and non-CTLD fragments of CD93 (specifically, the first two epidermal growth factor-like domains) alongside the N-terminal insulin-like growth factor-binding domain of IGFBP7 (PDB ID: 8IVD).
Figure 1. Structural representation of group XIV C-type lectin domain proteins. CD93 features an N-terminal signal peptide, a C-type lectin-like domain (CTLD) with conserved cysteine residues, a sushi-like domain (not present in Thrombomodulin), five epidermal growth factor (EGF)-like domains
(compared to six in Thrombomodulin, three in CD248, and one in CLEC14a), a mucin-like region with predicted O-linked glycosylation sites, a transmembrane region, and a cytoplasmic tail. And the structure of the CTLD and sushi domain of CD93 (CTLD-Sushi structural) has been confirmed by PDB deposition under the accession code 8A59.
Comment 2: The figures throughout the manuscript are of poor quality. Figure 1 lacks adequate labeling and explanation, and Figure 2 appears to include elements from BioRender, yet the overall visual style is inconsistent and resemble more of PowerPoint graphics.
Respond: We apologize for the initial image quality and have redrawn Figures 1-3 in response to the reviewer's comments. Instead of BioRender and PowerPoint, we utilized Figdraw, as noted in the Acknowledgments section.
“During the preparation of this manuscript, the authors used Figdraw for the purposes of drawing. “
Figure 2. The various binding ligands and interacting proteins of CD93. The extracellular domain of CD93 can bind to several ligands, including multimerin-2 (MMRN2), insulin-like growth factor binding protein 7 (IGFBP7), β-dystroglycan (DG), interleukin-17D (IL-17D) and vascular endothelial cadherin (VE- cadherin). And the region where CD93 and IGFBP7 bind has been confirmed by PDB deposition under the accession code 8IVD (this structure diagram is referenced from Protein Data Bank). The intracellular domain of CD93 can interact with the adaptor protein Cbl, moesin and GIPC. Binding of CD93 to β-DG leads to phosphorylation of tyrosine residues Tyr628 and Tyr644 in the cytoplasmic tail of CD93 by Src kinase, which stimulates the phosphorylation of the downstream effector Cbl. Phosphorylation of Cbl at Tyr774 may enable it to function as an adaptor protein, facilitating the recruitment of Crk and inducing
cell migration and tubular structure formation. Additionally, the interaction between CD93 and GIPC can mediate polarized transport through the branching of linear actin fiber bundles.
Figure 3. CD93 dysregulation in various diseases. In the context of clinical diseases, CD93 has been implicated in a range of pathological conditions, including inflammation (e.g., asthma, peritonitis, meningitis, periodontitis, etc.), cardiovascular diseases (e.g., atherosclerosis, diabetes, obesity, viral myocarditis (VMC), coronary heart disease (CHD), etc.), autoimmune disorders (e.g., rheumatoid arthritis (RA), systemic sclerosis (SSc), systemic lupus erythematosus (SLE), psoriasis, etc.), various cancers (e.g., stomach adenocarcinoma (STAD), colon adenocarcinoma (COAD), skin cutaneous melanoma (SKCM), nasopharyngeal carcinoma (NPC), liver hepatocellular carcinoma (LIHC), etc.), and other diseases (e.g., age-related macular degeneration (AMD), pre-eclampsia (PE), Parkinson's disease (PD), polycystic ovary syndrome (PCOS), etc.).
Comment 3: The statement that GST on CD93 allows it to bind to Moesin May be misleading and could confuse readers into thinking CD93 naturally contains a glutathione S-transferase domain.
Respond: We appreciate the reviewer’s comments and apologize for the confusion. We have substantially revised these sentences(lines 498-503).
“Researchers constructed glutathione S-transferase (GST) fusion proteins containing either the 47-amino acid cytoplasmic domain (Cyto) of GST or various mutants of the CD93 intracellular structural domain. The aim was to identify intracellular molecules that bind to CD93. When cell lysates or recombinant Moesin were used as a source of interacting molecules, the researchers found that the membrane proteins could bind to the GST-Cyto fusion protein.”
Comment 4: The section on GIPC is too brief and should be expanded to explain the functional consequences of CD93–GIPC interaction, such as its role in endocytosis and cytoskeletal dynamics. Respond: Thank you for the detailed review, we have made a substantial revision to this section(lines 573-584). “In 2005, Bohlson et al. conducted a yeast two-hybrid screen to identify proteins binding to the cytoplasmic tail of CD93. They identified Gα Interacting Protein (GAIP) Interacting Protein C-terminal (GIPC), which regulates migration in various systems and interacts with CD93 via a previously unrecognized CD93 class I PSD-95/Dlg/ZO-1 (PDZ) binding domain in C11. In mammalian cell lysates, the C11 domain and the highly charged near-membrane region of CD93 are essential for its effective interaction with GIPC. Upon stimulation, the C11 domain of CD93's cytoplasmic tail modulates phagocytic activity by interacting with PDZ domain-containing proteins. Additionally, GIPC interacts with myosin VI, α-actinin, and kinase family member KIF-1B. Both myosin VI and α-actinin are known to regulate endocytosis, highlighting GIPC's crucial role in this process. Another study found that overexpressing wild-type GIPC inhibited HUVEC migration in a damage assay.”
Comment 5: The IGFBP7 interaction deserves a more thorough molecular description, especially given the availability of structural data on pdb. The authors should detail the binding interface, the domains involved, and the implications for endothelial cell behavior and tumor vasculature.
Respond: Thank you very much for your valuable comments, we have made substantial changes to this section(lines 666-703). “The recently discovered glycoprotein IGFBP7, also known as IGFBP-rP1, AGM, T1A12, TAF, mac25, and PSF is a natural ligand of the CD93 receptor. IGFBP7, a 30 kDa member of the IGFBP family, features multiple conserved structural domains, including N-terminal, C-terminal, and central regions. The N-terminal domain harbors the IGFBP motif (GCGCCXXC), crucial for binding insulin-like growth factors (IGFs). IGFBP7 interacts with various molecules such as insulin, activin, and chemokines (e.g., CCL5, CCL21, CXCL10), as well as growth factors like VEGF-A and IGFs. Known receptors for IGFBP7 include the IGF-I receptor, integrin αvβ3, and CD93. Functioning as an extracellular matrix protein, IGFBP7 regulates fundamental biological processes such as proliferation, apoptosis, and migration, and significantly influences tumor development and angiogenesis. During physiological angiogenesis, IGFBP7 does not directly stimulate endothelial cell growth or migration, but rather supports efficient adhesion of endothelial cells, activates normal fibroblasts, and induces the expression of junction proteins to promote lumen formation. Moreover, IGFBP7 induces endothelial cell contraction through actin stress fibers and relaxes VE-cadherin-mediated intercellular junctions, thereby affecting vascular permeability and contributing to vascular stabilization and maturation. Upregulated IGFBP7 has also been observed in the vascular system of patients with traumatic brain injury and stroke, suggesting its involvement in vascular repair and remodeling processes, such as muscle hypertrophy, wound healing, and physiological and pathological fibrosis. Additionally,
the role of IGFBP7 in cancer has been extensively studied, with several reports confirming its overexpression in tumor vasculature and some human cancer cell lines. IGFBP7 exhibits a dual role in cancer, exerting antitumor effects by inhibiting tumor cell growth and accelerating apoptosis, while also inducing a disordered tumor vasculature system by binding to CD93. Sun et al. revealed that the interaction between CD93 and IGFBP7 is independent of IGF, relying instead on the CTLD structural domain and non-CTLD fragments of CD93 (specifically, the first two epidermal growth factor-like domains) alongside the N-terminal insulin-like growth factor-binding domain of IGFBP7 (PDB ID: 8IVD). This CD93-IGFBP7 interaction leads to aberrant tumor vascularity. By inhibiting their binding without altering vascular density, pericyte coverage is improved, promoting vascular maturation, reducing vascular leakage, decreasing tumor hypoxia, and enhancing tumor perfusion. These changes facilitate drug delivery and augment the anti-tumor efficacy of chemotherapy and immunotherapy. Additionally, the signal molecules associated with the VEGF and TGF-β pathways, influenced by IGFBP7, may be affected by the IGFBP7-CD93 interaction, although the precise signaling pathway requires further validation. In 2024, further research demonstrated that IGFBP7 and CD93 co-localize on the surface of human aortic endothelial cells (HAECs). IGFBP7 causes endothelial cell damage by interacting with CD93 and reducing SIRT1 expression through an autocrine mechanism.”
Comment 6: The discussion of CD93-targeting drugs is currently in the conclusion, which undermines its importance. This content should be integrated into the main body of the review and expanded into a dedicated section. It should cover therapeutic strategies such as monoclonal antibodies and small molecules, and discuss their mechanisms, efficacy, and clinical development status in more details.
Respond: We are very grateful for the constructive and insightful comments from the reviewers. We have included this part as the main body of the review and expanded it into a dedicated section(lines 4615-4632). As DCSZ11 entered Phase 2 clinical trials on July 15, 2025, I have also updated this in Table 2.
“Multiple experiments abroad have been conducted to develop CD93-related drugs, including DCBY 02, WO2024115637, CD34-Derived Allogeneic Dendritic Cell Cancer Vaccine, and DCSZ11. Concurrently, several research institutions in China have made similar efforts, such as SY2053, Anti CD93 antibody (China Resources Biopharmaceutical), CN116271112, and JS013 (as shown in Table 2). DCBY 02, Anti CD93 antibody (China Resources Biopharmaceutical), SY2053, and DCSZ11 are all CD93 monoclonal antibodies aimed at treating cancer by inhibiting CD93. The development of DCBY 02 has been terminated, and there has been no progress in the development of SY2053. The clinical development of Anti CD93 antibody (China Resources Biopharmaceutical) remains in the preclinical stage. However, DCSZ11 entered Phase 2 clinical trials (NCT07035249) on July 15, 2025, targeting metastatic solid tumors and head and neck squamous cell carcinoma. CN116271112 (Patent (CN116271112A)) is an antibody-conjugated radionuclide drug, and WO2024115637 (Patent (WO2024115637A1)) is a CD93 small molecule inhibitor, both of which
entered the drug discovery stage in 2023 and 2024, respectively. CD34-Derived Allogeneic Dendritic Cell Cancer Vaccine is a therapeutic vaccine that can treat not only tumors but also diseases related to the digestive system, endocrine, and metabolism, and it entered the preclinical stage on April 28, 2025, with a focus on pancreatic cancer (data are referenced from https://synapse.zhihuiya.com/).”
Comment 7: “Table 93”. line 56 page 2 i guess it should be “CD93”?
Respond: Thank you for your rigorous consideration, we regrettably made this error and have made appropriate modifications(lines 71-73).
“The CTLD family member CD93 is closely related to three other transmembrane glycoproteins: thrombomodulin (CD141, TM), CD248/endosialin (tumor endothelial marker 1/TEM1), and CLEC14a (as shown in Table 1).”
Thank you again for your valuable opinions and suggestions. We look forward to receiving your comments on what we have submitted and will be glad to respond to any questions or comments you may have.
Yours sincerely,
Menghan Cai

Reviewer 3 Report

Comments and Suggestions for Authors

The manuscript provides a comprehensive overview of CD93 in health and disease, with special attention to its role in cancer. While the review is generally informative, I have several concerns regarding the sections dealing with vasculogenic mimicry (VM) and the interactions between VE-cadherin and CD93:

  1. Vasculogenic Mimicry (VM): Insufficient Contextualization
    The authors briefly mention the implication of CD93 in vasculogenic mimicry; however, this important concept is underdeveloped and not sufficiently contextualized. The introduction lacks a proper historical and mechanistic overview of VM, including its origin, discovery in aggressive melanoma, and subsequent evidence across multiple tumor types. Without this background, readers unfamiliar with VM may not appreciate its significance in tumor biology or its relationship to CD93. I strongly recommend that the authors expand this section with key references that established VM as a hallmark of tumor plasticity and an alternative vascularization mechanism.

  2. Relationship Between VE-Cadherin and VM: Missing Depth
    The manuscript mentions VE-cadherin (VE-Cad) in the context of CD93 interactions, but it does not sufficiently emphasize the central role of VE-Cad in VM formation. A considerable body of work has shown that VE-Cad is not only an endothelial junctional protein but also a critical marker and functional regulator of VM in tumor cells. The current discussion in the manuscript is too superficial and does not adequately reflect the extensive evidence supporting the VE-Cad–VM axis. The authors should enrich this section with references to the numerous studies that have documented the essential role of VE-Cad in VM formation, cancer cell plasticity, and tumor progression.

  3. Link Between VE-Cad, CD93, and VM: Underdeveloped Argument
    Although CD93 is listed as an interacting partner of VE-Cad, the manuscript does not convincingly argue how this interaction could mechanistically relate to VM. Given the existing literature on VE-Cad as a driver of VM and the authors’ focus on CD93, this represents a missed opportunity. The authors should explicitly discuss potential molecular and signaling links between CD93 and VE-Cad in the context of VM, which could provide a more compelling narrative and novel perspective for readers.

The sections on VM and VE-cadherin require significant strengthening with additional background, mechanistic detail, and updated references. This would substantially improve the coherence and scientific depth of the manuscript.

Author Response

Dear Reviewer 3,
Thank you very much for taking the time to review this manuscript and we have studied your comments point by point, revised the manuscript accordingly.
To facilitate this discussion, we first retype your comments in bold and then highlight the changes in red in the revised manuscript.
Comment :
The sections on VM and VE-cadherin require significant strengthening with additional background, mechanistic detail, and updated references.
Respond:Thank you for the detailed review, we have made a substantial revision to this section (lines 736-770).
Since the introductory section emphasizes CD93's structural features. In "Interaction of CD93 with VE-cadherin," we detail VM's origin, outline the mechanism, discuss its presence in non-melanoma tumors, and examine its role in tumor progression. The discussion highlights VE-Cadherin's pivotal role in VM formation and related pathways, emphasizing their effects on cancer cell plasticity and tumor progression. Additionally, potential molecular and signaling connections between CD93 and VE-Cadherin in the VM context are clearly outlined.
“VE-cadherin is a transmembrane adhesion protein specifically expressed on the surface of vascular ECs. It primarily mediates cell-cell adhesion between ECs and is involved in the regulation of vascular endothelial growth factor (VEGF) receptor function, endothelial cell permeability, and vascular integrity. Therefore, VE-cadherin is closely associated with angiogenesis. Lugano et al. demonstrated that CD93 can interact with VE-cadherin and limit its phosphorylation and turnover. CD93 deficiency induces VE-cadherin phosphorylation under basal conditions, which disrupts endothelial cell connectivity. In conclusion, CD93 regulates the phosphorylation and turnover of VE-cadherin at endothelial junctions through a Rho/Rho kinase-dependent pathway, with implications for vascular stability and permeability. Additionally, VE-cadherin is a biomarker for vasculogenic mimicry (VM), a new model of tumor microcirculation. Unlike classical tumor angiogenesis, which depends on ECs, VM involves tumor cells exhibiting endothelial-like behavior and forming tubular structures to transport blood, allowing them to obtain oxygen and nutrients independently of normal blood vessels. VM represents a novel model of tumor microcirculation, distinct from classical tumor angiogenesis that relies on ECs. In VM, tumor cells mimic endothelial behavior, forming tubular structures to transport blood, thereby independently accessing oxygen and nutrients without relying on conventional vasculature. First introduced by Maniotis et al. in 1999 in human uveal melanoma, this phenomenon challenges the traditional view that endothelial cell-mediated angiogenesis is the sole mechanism for tumor growth and metastasis, while also complementing existing angiogenesis theories. Recently, VM has been identified in various malignancies, including glioblastoma, hepatocellular carcinoma, breast cancer, and lung cancer. VM correlates with tumor malignancy, growth, progression, metastasis, invasion, durg resistance, and poor prognosis. In 2001, Hendrix et al. demonstrated that reducing VE-cadherin expression in invasive melanoma cells disrupted VM network formation, directly linking VE-cadherin to VM. Similar findings were observed in a pancreatic cancer model. VE-cadherin primarily modulates the phosphorylation and
localization of embryonic stem cell markers, such as EphA2, to activate FAK and extracellular signal-regulated kinase (Erk). Independently, activated EphA2 can also initiate the PI3K/MMP pathway, promoting the formation of VM. On the other hand, CD93 induces VE-cadherin phosphorylation, disrupting endothelial cell junctions and the vascular barrier, leading to the leakage of plasma proteins (e.g., fibrinogen) and the formation of a fibrous network, which is a key factor in VM.Moreover, knockdown of CD93 and MMRN2 inhibits the activation of integrin β1, thereby suppressing the PI3K/AKT/SP2 signaling pathway and inhibiting VM formation.”
Thank you again for your valuable comments and suggestions. We look forward to hearing from you regarding our submission and we would be glad to respond to any further questions and comments that you may have.
Yours sincerely,
Menghan Cai
